# PAK-UCB Contextual Bandit: An Online Learning Approach to Prompt-Aware Selection of Generative Models and LLMs

Xiaoyan Hu [1]   Ho-fung Leung [2]   Farzan Farnia [1]

## Abstract

Selecting a sample generation scheme from multiple prompt-based generative models, including large language models (LLMs) and prompt-guided image and video generation models, is typically addressed by choosing the model that maximizes an averaged evaluation score. However, this score-based selection overlooks the possibility that different models achieve the best generation performance for different types of text prompts. An online identification of the best generation model for various input prompts can reduce the costs associated with querying sub-optimal models. In this work, we explore the possibility of varying rankings of text-based generative models for different text prompts and propose an online learning framework to predict the best data generation model for a given input prompt. The proposed PAK-UCB algorithm addresses a contextual bandit (CB) setting with shared context variables across the arms, utilizing the generated data to update kernel-based functions that predict the score of each model available for unseen text prompts. Additionally, we leverage random Fourier features (RFF) to accelerate the online learning process of PAK-UCB. Our numerical experiments on real and simulated text-to-image and image-to-text generative models show that RFF-UCB performs successfully in identifying the best generation model across different sample types. The code is available at: github.com/yannxiaoyanhu/dgm-online-select.

[1]Department of Computer Science and Engineering, The Chinese University of Hong Kong [2]Independent Researcher. Correspondence to: Xiaoyan Hu <xyhu21@cse.cuhk.edu.hk>, Farzan Farnia <farnia@cse.cuhk.edu.hk>.

*Proceedings of the $42^{nd}$ International Conference on Machine Learning*, Vancouver, Canada. PMLR 267, 2025. Copyright 2025 by the author(s).

## 1. Introduction

Large language models (LLMs) and text-guided generative AI models have found numerous applications in various engineering tasks. A prompt-guided generative AI represents a conditional generative model that produces output samples given an input text prompt. Over the past few years, several frameworks have been developed to train generative models and perform text-guided sample generation in various domains, including text, image, and video (Touvron et al., 2023; Gemini-Team et al., 2025; OpenAI et al., 2024; DeepSeek-AI et al., 2025; Rombach et al., 2022; Bao et al., 2023b; Podell et al., 2024; Singer et al., 2022; Bai et al., 2023; Liu et al., 2024). The great number of available LLMs and prompt-guided generative models has led to significant interest in developing evaluation mechanisms to rank the existing models and find the best model from a group of available ones. To address this task, several evaluation metrics have been proposed to quantify the fidelity of samples generated by prompt-based generative models, such as CLIPScore (Hessel et al., 2021) and PickScore (Kirstain et al., 2023) for prompt-guided image and video generation, and BLEU score (Papineni et al., 2002) and BERTscore (Zhang et al., 2020) for LLMs.

The existing model selection methodologies commonly aim to identify the generative model with the highest *averaged* fidelity score over the distribution of text prompts, producing samples that, on average, align the most with input text prompts. A well-known example is the averaged CLIPScore for text-to-image models, measuring the expected alignment between the input text and output image of the model using the CLIP embedding (Radford et al., 2021b). While the best-averaged score model selection strategy has been frequently utilized in generative AI applications, this approach does not consider the possibility that the involved models can perform differently across text prompts.

However, we highlight the possibility that one model outperforms another model in responding to text prompts from certain categories, while that model performs suboptimally in responding to prompts from other categories. Figure 1 shows examples of both LLMs and text-to-image models, where two widely-used models exhibit different rankings across the text prompts from different categories (code gen-

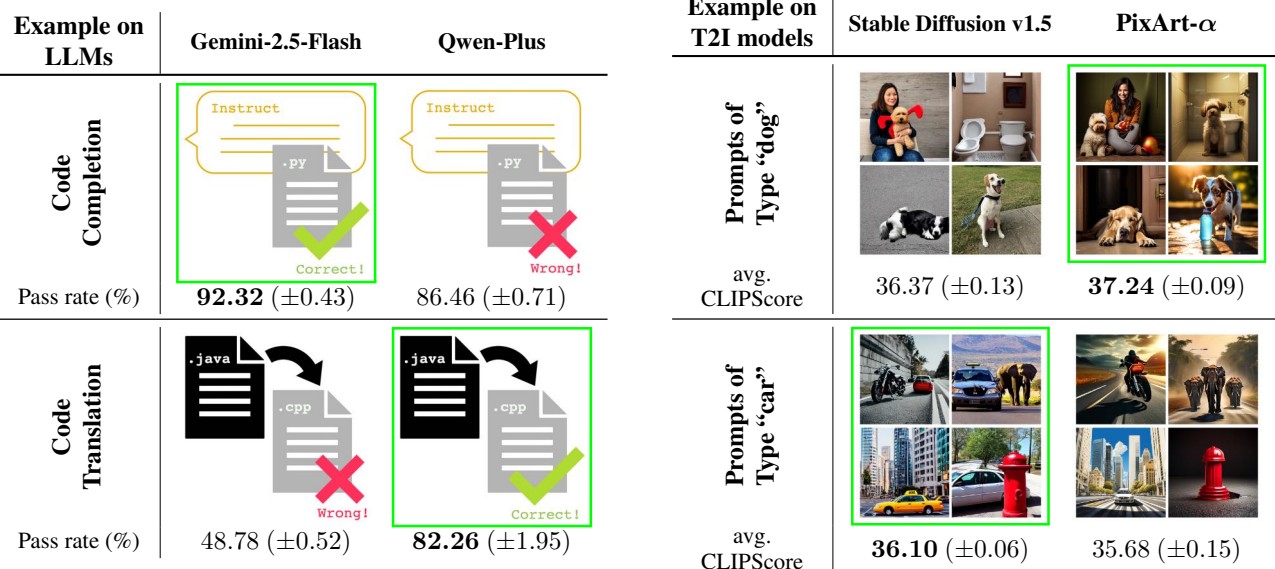

(a). Example 1: Gemini-2.5-Flash attains higher pass rate on (Python) code completion on the HumanEval benchmark (92.3% versus 86.5%) while underperforms for Java-to-C++ translation on the HumanEval-X benchmark (48.8% versus 82.3%).

(b). Example 2: Stable Diffusion v1.5 attains a higher CLIPScore in generating MS-COCO prompts with term "car" (36.10 versus 35.68) while underperforms for MS-COCO prompts with term "dog" (36.37 versus 37.24).

Figure 1. Widely-used LLMs and text-to-image models could exhibit different rankings across the text prompts with different categories.

eration and translation tasks for LLMs, and MS-COCO prompts with terms "dog"/"car"). In general, the different training sets and model architectures of LLMs and prompt-guided models can result in the models' varying performance in response to different text prompts, which is an important consideration in assigning prompts of various types to a group of prompt-input generative models.[1]

In this work, we aim to develop a learning algorithm to identify the best generative model for a given input prompt, using observed prompt/generated samples collected from the models in the previous sample generation rounds (Figure 2). Since the goal of prompt-based model selection is to avoid sample generation queries from suboptimal generative models for a given text prompt, we view the model selection task as an online learning problem, where after each data generation, the learner updates the prediction on which generative model performs the best in response to input text prompts. Here, the objective of the online learner is to utilize the previously generated samples to accurately guess the generation model with the best performance for the incoming text prompt. An optimal online model selection method will result in a bounded regret value, measured in comparison to the sample generation from the groundtruth-best model for the text prompts.

We highlight that the described online learning task can be viewed as a *contextual bandit (CB)* problem widely studied in the bandit literature (Langford & Zhang, 2007; Li et al., 2010; Chu et al., 2011; Agrawal & Goyal, 2013). In the CB task, the online learner observes a single context variable (the text prompt in our setting) and predicts the best arm for the current input context. Specifically, we focus on the kernel-based method to predict the score of each available model for an incoming prompt. As the text prompt is shared across the models, the CB task may be suboptimally addressed by the common-weight formulation of linear CB (Chu et al., 2011) and kernelized CB (Valko et al., 2013) algorithms, as these implementations of the algorithms learn a single shared weight vector to predict the reward functions for all the arms (i.e., generative models in our setting). To relax this constraint, we propose Per-Arm Kernelized UCB (PAK-UCB) to address the online prompt-based selection of generative models. According to the PAK-UCB approach, the learner utilizes the computed UCB-scores of arm-specific kernel-based prediction functions to choose the generative model for the incoming text prompt and subsequently update the kernel-based prediction rule based on the generated data for the future rounds. We attempt to theoretically analyze the proposed PAK-UCB and show that its variant could achieve a regret bound of $\widetilde{O}(\sqrt{T})$ over $T$ rounds.

Since the user applying the CB-based model selection ap-

---

[1]The full models' response codes in the figure's experiment are provided at: https://github.com/yannxiaoyanhu/dgm-online-select.

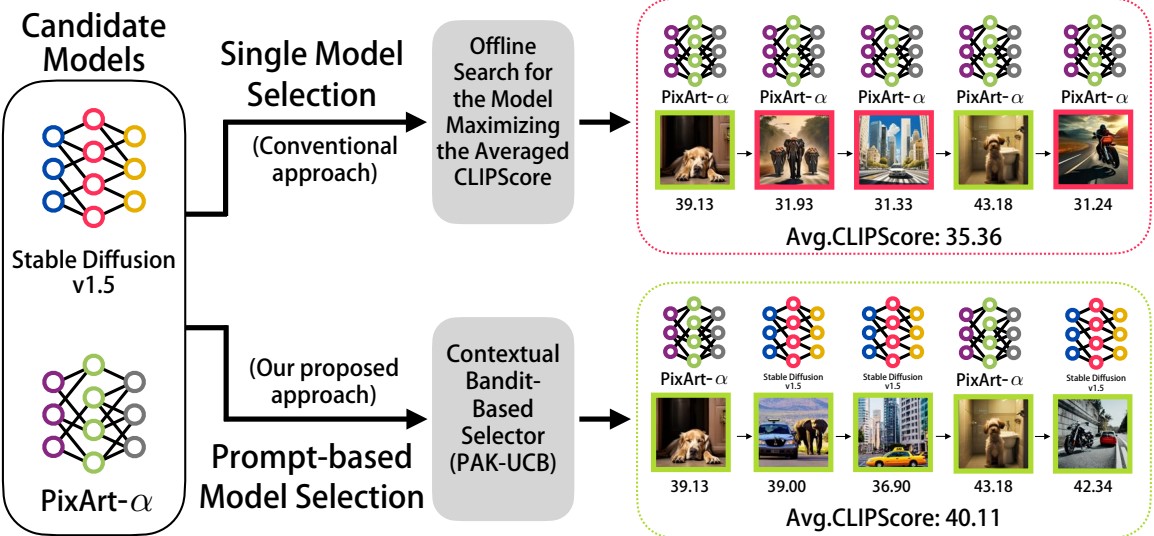

*Figure 2.* Selection of text-to-image (T2I) models based on the CLIPScore: The conventional best-score model selection assigns all prompts to the single model with the best averaged CLIPScore. In contrast, our proposed online approach performs a prompt-level model selection by leveraging the UCB contextual bandit algorithm to assign the model to the prompt, learning from previous data generations.

proach may have limited compute power and not be able to afford growing computational costs in the online learning process, we propose applying the random Fourier features (RFF) framework (Rahimi & Recht, 2007b) to reduce the computational load of PAK-UCB. We discuss that in the proposed PAK-UCB, the computational cost per iteration will grow cubically as $O(t^3)$ with iteration $t$. We demonstrate that after leveraging the RFF method, our developed proxy RFF-UCB algorithm can effectively approximate the solution to PAK-UCB with the computational costs increasing only linearly $O(t)$.

Finally, we present the results of several numerical experiments to show the efficacy of our proposed PAK-UCB and RFF-UCB in the online selection of prompt-guided generative models and LLMs. In our experiments, we test several pre-trained and simulated text-to-image, image-captioning (image-to-text), and language models, where different models could lead to different rankings across various sample types. Our numerical results suggest a fast convergence of the proposed online learning algorithms to the best prompt-based selection of the available models in response to different prompt types. In our experiments, the proposed PAK-UCB and RFF-UCB could perform better than several baseline methods, including the common-weight implementation of LinUCB (Chu et al., 2011) and KernelUCB (Valko et al., 2013) methods. The following is a summary of this work's contributions:

- Studying the prompt-based selection of prompt-guided generative models to improve the prompt-level performance scores,

- Developing the contextual bandit algorithms of PAK-UCB and RFF-UCB algorithms for the online selection of prompt-based generative models,

- Theoretically analyzing the regret and computational costs of PAK-UCB and RFF-UCB online learning methods

- Presenting numerical results on the prompt-based selection of generative models using PAK-UCB and RFF-UCB.

## 2. Related Work

**(Automatic) Evaluation of conditional generative models.** Evaluating the conditional generative models has been studied extensively in the literature. For text-to-image (T2I) generation, earlier methods primarily rely on the Inception score (Salimans et al., 2016) and Fréchet inception distance (Heusel et al., 2017). More recent works propose reference-free metrics for robust automatic evaluation of T2I and image captioning, with notable examples being CLIP-Score (Hessel et al., 2021) and PickScore (Kirstain et al., 2023). Kim et al. (2022) propose a mutual-information-based metric, which attains consistency across benchmarks, sample parsimony, and robustness. To provide a holistic evaluation of T2I models, several works focus on multi-objective evaluation. Astolfi et al. (2024) propose to evaluate conditional image generation in terms of *prompt-sample consistency*, *sample diversity*, and *fidelity*. Kannen et al. (2024) introduce a framework to evaluate T2I models regarding *cultural awareness* and *cultural diversity*. Masrourisaadat et al. (2024) examine the performance of several T2I models in generating images such as human faces and

groups and present a social bias analysis. Another line of study explores evaluation approaches using large language models (LLMs). Tan et al. (2024) develop LLM-based evaluation protocols that focus on the *faithfulness* and *text-image alignment*. Peng et al. (2024) introduce a GPT-based benchmark for evaluating personalized image generation. In addition, a line of study leverages human feedback for scoring/ranking generated images and improving T2I models (Xu et al., 2023). For evaluation of text-to-video (T2V) generation, Huang et al. (2024) introduce VBench as a comprehensive evaluation of T2V models in terms of *quality* and *consistency*. Finally, we note the prompt-aware diversity scores, Conditional-Vendi (Jalali et al., 2023) and Schur-Complement-Entropy (Ospanov et al., 2024a), developed for prompt-guided generative models.

**(Kernelized) Contextual bandits.** The contextual bandits (CB) is an efficient framework for online decision-making with side information (Langford & Zhang, 2007; Foster et al., 2018), which is widely adopted in domains such as recommendation system and online advertisement (Li et al., 2010) and resource allocations (Lim et al., 2024). A key to its formulation is the relationship between the context (vector) and the expected reward. In linear CB, the reward is assumed to be linear to the context vector (Li et al., 2010; Chu et al., 2011). To incorporate non-linearity, Valko et al. (2013) propose kernelized CB, which assumes the rewards are linear-realizable in a reproducing kernel Hilbert space (RKHS). To address the growing computation, recent work leverages the assumption that the kernel matrix is often approximately low-rank and uses Nyström approximations (Calandriello et al., 2019; 2020; Zenati et al., 2022).

**Prompt-based Selection of LLMs and Generative Models.** We note that a line of study proposes offline methods for learning a universal prompt-to-model assignment rule by training a neural network model over a dataset of ranked responses of the models to a large set of prompts (Luo et al., 2024; Qin et al., 2024; Frick et al., 2025). Unlike these methods, our proposed approach follows an online learning strategy which can significantly lower the computational and statistical costs of finding a satisfactory assignment rule. In addition, as we later discuss in the numerical results, the online nature of our proposed algorithm can improve the adaptability of the model to the changes in the user's prompt distribution and models' behavior over the iterations of evaluating the models in real time.

**Multi-armed bandit selection of generative models.** We note that the references (Hu et al., 2025a; Rezaei et al., 2025) propose multi-armed bandit (MAB) approaches to the selection of unconditional generative models without an input text prompt. Specifically, Hu et al. (2025a) propose an online learning framework for selecting generative models and develop the FID-UCB algorithm for the online selection of image-output generative models based on the FID score (Heusel et al., 2017). Rezaei et al. (2025) show that a mixture of several unconditional generative models can attain higher diversity scores than each individual model, and propose the Mixture-UCB MAB framework to find the best mixture of the generative models in terms of the kernel distance (Bińkowski et al., 2018) and Renyi kernel entropy (Jalali et al., 2023; Ospanov et al., 2024b). We note that the mentioned papers study the selection of unconditional generative models, different from our problem setting of selecting the conditional prompt-guided generative models.

## 3. Preliminaries

### 3.1. Kernel Methods and Random Fourier Features

Let $\phi : \mathbb{R}^d \to \mathcal{H}$ be a feature map from the *ambient space* $\mathbb{R}^d$ to the (possibly infinite-dimensional) space corresponding to the associated *reproducing kernel Hilbert space* (RKHS) $\mathcal{H}$. The *kernel function* for RKHS $\mathcal{H}$ is defined as the inner product of the representations of the inputs as $k(y, y') := \langle \phi(y), \phi(y') \rangle = \phi(y)^\top \phi(y')$ for every $y, y' \in \mathbb{R}^d$, where $\langle \cdot, \cdot \rangle$ denotes the standard inner product.

Every kernel function $k$ will satisfy the positive-definiteness condition, that means $\sum_{i=1}^n \sum_{j=1}^n c_i c_j k(y_i, y_j) \geq 0$ holds for every integer $n \in \mathbb{N}_+$, $y_1, \cdots, y_n \in \mathbb{R}^d$, and $c_1, \cdots, c_n \in \mathbb{R}$. Furthermore, we call a kernel function $k$ *shift invariant* if $k(y, y') := k(y - y')$ for every $y, y' \in \mathbb{R}^d$. A well-known example is the *radial basis function* (RBF) kernel with bandwidth parameter $\sigma > 0$:

$$k_{\text{RBF}}(y, y') = \exp\left(\frac{-\|y - y'\|_2^2}{2\sigma^2}\right).$$

**Kernel ridge regression (KRR).** Given empirical labeled samples $(y_1, s_1), \cdots, (y_n, s_n)$, where $\{y_i \in \mathbb{R}^d\}_{i=1}^n$ are *feature variables* and $\{s_i \in \mathbb{R}\}_{i=1}^n$ are *target variables*, respectively, the kernel ridge regression assumes that for a $w^\star \in \mathcal{H}$ we have $\mathbb{E}[s_i|y_i] = \phi(y)^\top w^\star$ for any $i = 1, \cdots, n$. For regularization parameter $\alpha \geq 0$, KRR estimator is

$$\widehat{s}_{\text{KRR}}(y) := k_y^\top (K + \alpha I_n)^{-1} v$$

for every $y \in \mathbb{R}^d$, where $K = [k(y_i, y_j)]_{i,j=1}^n \in \mathbb{R}^{n \times n}$ denotes the kernel matrix, $v := [s_1, \cdots, s_n]^\top \in \mathbb{R}^n$, and $k_y = [k(y_1, y), \cdots, k(y_n, y)]^\top \in \mathbb{R}^n$. The KRR estimator can be interpreted as the solution $w^\star$ to the ridge regression:

$$\widehat{w} := \arg\min_{w \in \mathcal{H}} \sum_{i=1}^n \left(\phi(y_i)^\top w - s_i\right)^2 + \alpha \|w\|^2,$$

where $\|w\| := \sqrt{w^\top w}$ for any $w \in \mathcal{H}$, and then making the prediction $\widehat{s}_{\text{KRR}}(y) = (\phi(y))^\top \widehat{w}$.

**Random Fourier features (RFF).** To reduce the computational costs of kernel methods for shift-invariant kernels,

Rahimi & Recht (2007a) proposes the random Fourier features framework. Specifically, Bochner's Theorem (Rudin, 2017) shows that for every shift-invariant kernel $k(y, y') = \kappa(y-y')$ satisfying $\kappa(0) = 1$, the Fourier transform of $\kappa$ provides a valid probability density function, which we denote by $p \in \Delta(\mathbb{R}^d)$, such that $k(y, y') = \mathbb{E}_{w \sim p}[e^{iw^\top (y-y')}]$, with $i$ being the imaginary unit. Following this property, the RFF approach independently samples $w_1, \cdots, w_D \sim p$ and proposes the empirical mean proxy for $k(y, y')$ as $\frac{1}{D}\sum_{j=1}^{D} e^{iw_j^\top (y-y')}$. Due to the real Fourier transform of the even function $\kappa$, we can simplify the equation as

$$\varphi(y) = \frac{1}{\sqrt{D}}\big[\cos(w_1^\top y), \sin(w_1^\top y), ., \cos(w_D^\top y), \sin(w_D^\top y)\big] \quad (1)$$

where $\{w_j\}_{j=1}^{D} \overset{\text{i.i.d}}{\sim} p$ and $k(y, y')$ is approximated by the inner product $\varphi(y)^\top \varphi(y')$. The resulting approximate KRR estimator is

$$\widetilde{s}_{\text{KRR}}(y) := (\widetilde{\Phi}^* \widetilde{\Phi} + \alpha I_{2D})^{-1} \widetilde{\Phi}^* v,$$

where $\widetilde{\Phi} := [\varphi(y_i)^\top]_{i=1}^{n} \in \mathbb{R}^{n \times 2D}$ and we denote by $\widetilde{\Phi}^*$ its conjugate transpose, can be computed in $O(nD^2)$ time and $O(nD)$ memory, giving substantial computational savings if $D \ll n$. For the RBF kernel $k_{\text{RBF}}^{\sigma}(x_1, x_2) = \exp(-\frac{1}{2\sigma^2}\|x_1 - x_2\|_2^2)$, the PDF $p_{\text{RBF}}$ will be the multivariate Gaussian $\mathcal{N}(0, \frac{1}{\sigma^2} \cdot I_d)$.[2]

### 3.2. CLIPScore for Evaluating Text-to-Image Models

CLIPScore (Hessel et al., 2021) has been widely-used to evaluate the alignment of samples generated by text-to-image/video (T2I/V) and image captioning models. Let $(y, x) \in \mathcal{Y} \times \mathcal{X}$ be a *text-image pair*. We denote by $\boldsymbol{c}_y \in \mathbb{S}^{d-1} := \{z \in \mathbb{R}^d : \|z\|_2 = 1\}$ and $\boldsymbol{v}_x \in \mathbb{S}^{d-1}$ the (normalized) embeddings of text $y \in \mathcal{Y}$ and image $x \in \mathcal{X}$, respectively, both extracted by CLIP (Radford et al., 2021a). The CLIPScore (Hessel et al., 2021) is given by

$$\text{CLIPScore}^{\text{T2I}}(y, x) := \max\{0, 100 \cdot \cos(\boldsymbol{v}_x, \boldsymbol{c}_y)\}, \quad (2)$$

where $\cos(\boldsymbol{v}_x, \boldsymbol{c}_y) = \langle \boldsymbol{v}_x, \boldsymbol{c}_y \rangle$ for the $\ell_2$-normalized CLIP-embedded $\boldsymbol{v}_x, \boldsymbol{c}_y$. Extending the definition to (text,video) pairs, for a video $X := \{x^{(l)}\}_{l=1}^{L}$ consisting of $L$ frames, where $x^{(l)}$ is the $l$-th frame, the score is the averaged frame-level CLIPScore:

$$\text{CLIPScore}^{\text{T2V}}(y, X) := \frac{1}{L}\sum_{l=1}^{L} \text{CLIPScore}^{\text{T2I}}(y, x^{(l)}).$$

---

[2]We note that an alternative form of RFF is given by: $\varphi(y) = \sqrt{\frac{2}{D}} \cdot [\cos(w_1^\top y + b_1), \cdots, \cos(w_D^\top y + b_D)]^\top$, where $\{w_j\}_{j=1}^{D} \overset{\text{i.i.d}}{\sim} p$ and $\{b_j\}_{j=1}^{D} \overset{\text{i.i.d}}{\sim} \text{Unif}([0, 2\pi])$. For the RBF kernel, it can be shown that (1) has a uniformly lower variance (Sutherland & Schneider, 2015).

## 4. Prompt-Based Model Selection as a Contextual Bandit Problem

In this section, we introduce the framework of online prompt-based selection of generative models, which is given in Protocol 1. Let $[N] := \{1, \cdots, N\}$ for any positive integer $N \in \mathbb{N}_+$. We denote by $\mathcal{G} := [G]$ the set of (prompt-based) generative models. The evaluation proceeds in $T \in \mathbb{N}_+$ iterations.

At each iteration $t \in [T]$, a *prompt* $y_t \in \mathcal{Y}$ is drawn from a fixed distribution $\rho \in \Delta(\mathcal{Y})$ on the prompt space $\mathcal{Y} \subseteq \mathbb{S}^{d-1}$, e.g., (the normalized embedding of) a picture in image captioning or a paragraph in text-to-image/video generation. Based on prompt $y_t$ (and previous observation sequence), an algorithm $\mathcal{A}$ picks model $g_t \in \mathcal{G}$ and samples *response* $x_t \sim P_{g_t}(\cdot|y_t)$, where $P_g(\cdot|y) \in \Delta(\mathcal{X})$ is the conditional distribution of the generated response from any model $g \in \mathcal{G}$. The quality of response $x_t$ is evaluated using a *score function* $s : \mathcal{Y} \times \mathcal{X} \to [-1, 1]$, which assigns a score $s(y_t, x_t)$. The algorithm $\mathcal{A}$ aims to minimize the *regret*

$$\text{Regret}(T) := \sum_{t=1}^{T} (s_\star(y_t) - s_{g_t}(y_t)), \quad (3)$$

where we denote by $s_g(y) := \mathbb{E}_{x_g \sim P_g(\cdot|y)}[s(y, x_g)]$ the expected score of any model $g \in \mathcal{G}$ and $s_\star(y) := \max_{g \in \mathcal{G}} s_g(y)$ the optimal expected score, both conditioned on the prompt $y$.

---

**Protocol 1** Online Prompt-based Selection of LLMs and Prompt-guided Generative Models

---

**Require:** total iterations $T \in \mathbb{N}_+$, set of generators $\mathcal{G} = [G]$, prompt distribution $\rho \in \Delta(\mathcal{Y})$, score function $s : \mathcal{Y} \times \mathcal{X} \to [-1, 1]$, algorithm $\mathcal{A} : (\mathcal{Y} \times \mathcal{G} \times \mathbb{R})^* \times \mathcal{Y} \to \Delta(\mathcal{G})$
**Initialize:** observation sequence $\mathcal{D} \leftarrow \varnothing$
1: **for** iteration $t = 1, 2, \cdots, T$ **do**
2:     Prompt $y_t \sim \rho$ is revealed.
3:     Algorithm $\mathcal{A}$ picks model $g_t \sim \mathcal{A}(\cdot|\mathcal{D}, y_t)$ and samples response $x_t \sim P_{g_t}(\cdot|y_t)$.
4:     Score $s_t \leftarrow s(y_t, x_t)$ is assigned.
5:     Update observation $\mathcal{D} \leftarrow \mathcal{D} \cup \{(y_t, g_t, s_t)\}$.
6: **end for**

---

## 5. An Optimism-based Approach for Prompt-based Selection

Under the online prompt-based selection setting, a key challenge is to learn the relationship between the prompt and the score of each model. In this paper, we consider kernel methods for score prediction. Specifically, for each model $g \in \mathcal{G}$, we assume the existence of a (possibly non-linear and infinite-dimensional) feature map $\phi$ and a weight $w_g^\star$

in an RKHS, such that the score $s_g(y)$ conditioned on the prompt $y$ is given by $(\phi(y))^\top w_g^\star$. Notably, the weight vector $w_g^\star$ is arm-specific and can vary across the models, which is stated in the following assumption.

**Assumption 1** (Realizability). *There exists a mapping $\phi : \mathbb{R}^d \to \mathcal{H}$ and weight $w_g^\star \in \mathcal{H}$ such that score $s_g(y) = \langle y, w_g^\star \rangle_\mathcal{H}$ for any prompt vector $y \in \mathbb{R}^d$ and model $g \in \mathcal{G}$. Further, it holds that $\|w_g^\star\| \leq 1$, and $k(y, y) \leq \kappa^2$ and $\|\phi(y)\| \leq 1$ for any $y \in \mathcal{Y}$, where $k : \mathbb{R}^d \times \mathbb{R}^d \to \mathbb{R}$ is the kernel function of the mapping $\phi$.*

**Remark 1** (Comparison to linear and kernelized CB). *We note that Assumption 1 differs from the standard settings of linear and kernelized contextual bandit (CB) (Chu et al., 2011; Valko et al., 2013; Zenati et al., 2022). Specifically, in the settings of these references, a potentially different context variable is considered for each arm, and the same weight vector is applied across all arms. In the prompt-based model selection setting, the text prompt (i.e., the context variable) is shared among arms (i.e., the generative models) and can result in different performance scores across the models, which cannot be captured by the standard kernelized CB approach. On the other hand, we note that the existing arm-specific bandit algorithms in (Li et al., 2010; Xu et al., 2020) are designed considering a linear pay-off structure, which is generally different from the kernel-based formulation in our work.*

### 5.1. The PAK-UCB Algorithm

In this section, we present PAK-UCB in Algorithm 2, an online learning approach to prompt-based model selection. For each incoming prompt, PAK-UCB first estimates the performance scores via kernel ridge regression (KRR) and then picks the model with the highest estimated score. Unlike conventional formulations of LinUCB (Chu et al., 2011) and KernelUCB (Valko et al., 2013) algorithms, which learn a single reward function shared across all arms, PAK-UCB learns arm-specific functions to predict the score of each model.

The key component in PAK-UCB is the function COMPUTE_UCB (lines 8-17), which outputs both the KRR estimator $\widehat{\mu}_g$ and an uncertainty quantifier $\widehat{\sigma}_g$. As the weight vector $w_g^\star$ can vary across the arms, PAK-UCB constructs the KRR dataset using prompt-score pairs from iterations where model $g$ is chosen, with the corresponding indices stored in the set $\Psi_g$ (line 6). The estimated score is then computed by $\widehat{s}_g = \widehat{\mu}_g + \eta\widehat{\sigma}_g$ (line 4), which is initially set to $+\infty$ to ensure each model is picked at least once (lines 9-10). To provide theoretical justification for our method, we show that a variant of PAK-UCB, which is illustrated in Algorithm 3 in the Appendix, can attain a squared-root regret bound. The formal statement and the proof can be found in Appendix A.

---

**Algorithm 2** Per-Arm Kernelized UCB (PAK-UCB)

---

**Require:** iteration $T$, generators $\mathcal{G} = [G]$, prompt distribution $\rho$, score function $s : \mathcal{Y} \times \mathcal{X} \to [-1, 1]$, positive definite kernel $k$, regularization and exploration parameters $\alpha, \eta \geq 0$

**Initialize:** observation sequence $\mathcal{D} \leftarrow \varnothing$ and index set $\Psi_g \leftarrow \varnothing$ for all $g \in \mathcal{G}$

1: **for** iteration $t = 1, 2, \cdots, T$ **do**
2:     Prompt $y_t \sim \rho$ is revealed.
3:     Compute $(\widehat{\mu}_g, \widehat{\sigma}_g) \leftarrow$ COMPUTE_UCB$(\mathcal{D}, y_t, \Psi_g)$ and set $\widehat{s}_g \leftarrow \widehat{\mu}_g + \eta\widehat{\sigma}_g$ for each $g \in \mathcal{G}$.
4:     Pick model $g_t \leftarrow \arg\max_{g \in \mathcal{G}}\{\widehat{s}_g\}$.
5:     Sample $x_t \sim P_{g_t}(\cdot|y_t)$ and receive score $s_t$.
6:     Update $\mathcal{D} \leftarrow \mathcal{D} \cup \{(y_t, s_t)\}$ and $\Psi_{g_t} \leftarrow \Psi_{g_t} \cup \{t\}$.
7: **end for**

8: **function** COMPUTE_UCB$(\mathcal{D}, y, \Psi_g)$
9:     **if** $\Psi_g$ is empty **then**
10:       $\widehat{\mu}_g \leftarrow +\infty, \widehat{\sigma}_g \leftarrow +\infty$.
11:     **else**
12:       Set $K \leftarrow [k(y_i, y_j)]_{i,j \in \Psi_g}, v \leftarrow [s_i]_{i \in \Psi_g}^\top$, and $k_y \leftarrow [k(y, y_i)]_{i \in \Psi_g}^\top$.
13:       $\widehat{\mu}_g \leftarrow k_y^\top(K + \alpha I)^{-1}v$.
14:       $\widehat{\sigma}_g \leftarrow \alpha^{-\frac{1}{2}}\sqrt{k(y, y) - k_y^\top(K + \alpha I)^{-1}k_y}$.
15:     **end if**
16:     **return** $(\widehat{\mu}_g, \widehat{\sigma}_g)$.
17: **end function**

---

**Theorem 1** (Regret, informal). *With probability of at least $1 - \delta$, the regret of running a variant of PAK-UCB (stated in Algorithm 3 in the Appendix) for $T$ iterations is bounded by $\widetilde{O}(\sqrt{GT})$.*

### 5.2. PAK-UCB with Random Fourier Features

The PAK-UCB solves a KRR for each model at an iteration to estimate the scores, which can be expensive in both computation and memory for a large number of iterations. To address this problem, we leverage the random Fourier features (RFF) sampling (Rahimi & Recht, 2007a) for shift-invariant kernel functions, e.g., the RBF (Gaussian) kernel. At a high level, RFF maps the input data, e.g., the prompt (vector) in our setting, to a randomized low-dimensional feature space and then learns a linear model in the resulting random space by solving a standard linear regression problem. Note that following the design of RFF, the inner product of the projected randomized features will be an unbiased estimation of the original kernel inner product.

We present RFF-UCB as an RFF implementation of PAK-UCB (Algorithm 4 in the Appendix). Particularly, RFF-UCB leverages an RFF-based approach to compute the mean

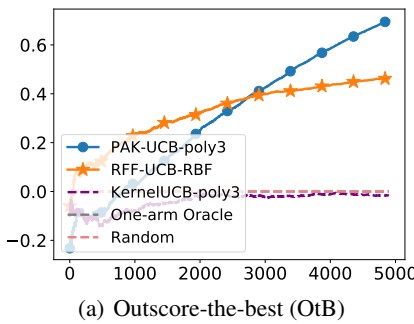

(a) Outscore-the-best (OtB)

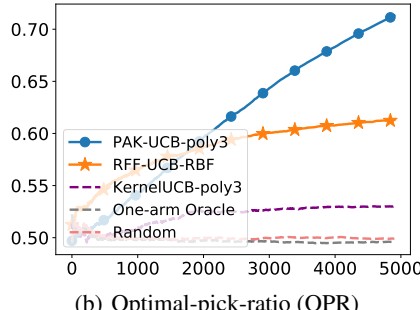

(b) Optimal-pick-ratio (OPR)

*Figure 3.* Prompt-based selection between Stable Diffusion v1.5 and PixArt-$\alpha$ (Figure 1(a)): Results are averaged over 20 trials.

and uncertainty quantifier, which is stated in Algorithm 5 in the Appendix. Upon receiving the regression dataset consisting of prompt-score pairs, COMPUTE_UCB_RFF first projects each $d$-dimensional prompt vector to a randomized $2D$-dimensional feature space according to Equation (1) and then solves a linear ridge regression to estimate the mean and uncertainty. To see why RFF can reduce the computation, note that the size of the (regularized) Gram matrix $(\widetilde{\Phi}_g^\top \widetilde{\Phi}_g + \alpha I_{2D})$ in Line 8 is fixed to be $2D$ in the whole process, while the size of $(K + \alpha I)$ in line 13 of Algorithm 2 scales with $|\Psi_g|$ and can grow linearly over iterations. Particularly, the following lemma shows that COMPUTE_UCB_RFF can reduce the time and space by an order of $O(t^2)$ and $O(t)$, respectively.

**Lemma 1** (Time and space complexity). *At any iteration $t \in [T]$, COMPUTE_UCB (Lines 8-17 of Algorithm 2) requires $O(t^3/G^2)$ time and $O(t^2/G)$ space, while COMPUTE_UCB_RFF with random features of size $s \in \mathbb{N}_+$ (Algorithm 5) requires $O(tD^2)$ time and $O(tD)$ space, where $G$ is the number of generators. See Appendix B.1 for details.*

To provide theoretical justification to the RFF approach, we show that the implementation of PAK-UCB with RFF can attain the exact same regret bound with a carefully chosen feature sizes. The formal statement and the proof can be found in Appendix B.1.

**Theorem 2** (Regret of RFF implementation, informal). *With probability at least $1 - \delta$, the regret of running a variant of RFF-UCB (stated in Algorithm 6) for $T$ iterations is bounded by $\widetilde{O}(\sqrt{GT})$.*

## 6. Numerical Results

In this section, we present numerical results for the proposed 1) PAK-UCB-poly3: PAK-UCB using a polynomial kernel with degree 3, i.e., $k_{\text{poly3}}^\gamma(x_1, x_2) = (1 + \gamma \cdot x_1^\top x_2)^3$ and 2) RFF-UCB-RBF: PAK-UCB with RFF implementation of the RBF kernel, i.e., $k_{\text{RBF}}^\sigma(x_1, x_2) = \exp(-\frac{1}{2\sigma^2}\|x_1 - x_2\|^2)$. Implementation details and the choice for hyperparameters can be found in Appendix C.4. Our primary focus is on

prompt-based selection among standard text-to-image (T2I) models: Stable Diffusion v1.5[3], PixArt-$\alpha$-XL-2-512x512[4], UniDiffuser[5], and DeepFloyd IF-I-M-v1.0.[6]. For the LLM experiments, we provide numerical results for prompt-based selection of the following large language models (LLMs): Gemini-2.5-Flash-preview, o3-mini, Deepseek-Chat, and Qwen-Plus. In the Appendix, we report additional results on image captioning (image-to-text) task and video data under synthetic setups. Additional results can be found in Appendix C.

**Baselines.** We compare PAK-UCB-poly3 and RFF-UCB-RBF with three baselines, including 1) KernelUCB-poly3: standard KernelUCB (Valko et al., 2013) using the $k_{\text{poly}}^{1.0}$ kernel, 2) One-arm Oracle: always picking the model with the maximum averaged score, and 3) Random: selecting an model uniformly randomly. In the Appendix, we report results for three additional baselines: 4) PAK-UCB-lin: PAK-UCB with linear kernel, i.e., $k_{\text{lin}}(x_1, x_2) = x_1^\top x_2$, which does not incorporate non-linearity in score estimation, 5) LinUCB: standard LinUCB (Chu et al., 2011), and 6) Naive-KRR: PAK-UCB-poly3 without exploration, i.e., parameter $\eta = 0$, which selects the model with the highest estimated mean conditioned to the prompt.

**Performance metrics.** For each experiment, we report two performance metrics: (i) *outscore-the-best* (OtB): the difference between the CLIPScore attained by the algorithm and the highest average CLIPScore attained by any single model, and (ii) *optimal-pick-ratio* (OPR): the overall ratio that the algorithm picks the best generator conditioned to the prompt. Specifically, to determine the best model for each specific prompt, we generate five responses from every model (to that prompt), and the best model in the OPR calculation is the model with the highest averaged score

---

[3] https://huggingface.co/docs/diffusers/en/api/pipelines/stable_diffusion/text2img
[4] https://huggingface.co/PixArt-alpha/PixArt-XL-2-512x512
[5] https://github.com/thu-ml/unidiffuser
[6] https://github.com/deep-floyd/IF

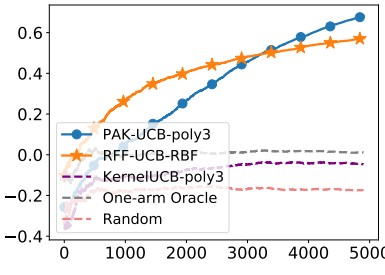 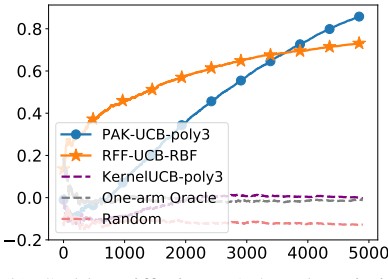 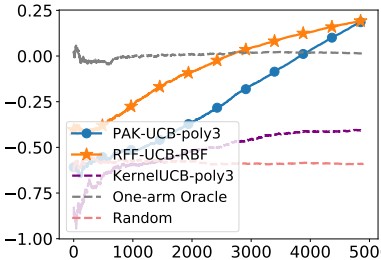

(a) UniDiffuser and PixArt-$\alpha$ (Figure 9).

(b) Stable Diffusion v1.5 and UniDiffuser (Figure 11)

(c) Stable Diffusion v1.5, PixArt-$\alpha$, and DeepFloyd

*Figure 4.* Prompt-based selection among several standard T2I models: Outscore-the-best (OtB) is reported. The selection of our proposed PAK-UCB-poly3 algorithm converges quickly to the optimal model for incoming text prompts. Full results can be found in Figures 10, 12, and 13 in the Appendix. Results are averaged over 20 trials.

over the 5 runs. Then, OPR is defined as the fraction that the algorithm's selected model matches the best model.

### 6.1. Summary of the Numerical Results

The results of our numerical experiments indicate the improvement of the proposed PAK-UCB algorithm over the one-arm oracle baseline that is aware of the single generative model with the maximized averaged scores over the prompt distribution. This result suggests that the online learning algorithm could outperform a selector with the side-knowledge of the single best-performing model. We note that this improvement follows the *prompt-based selection* design of our proposed PAK-UCB algorithm. Moreover, our numerical results indicate that PAK-UCB can significantly improve upon the online learning baselines with shared weights, including LinUCB (Chu et al., 2011) and KernelUCB (Valko et al., 2013) methods, indicating the effectiveness of the arm-specific design of PAK-UCB in the prompt-based model selection setting. Finally, in our experiments, the proposed RFF-UCB-RBF variant could reduce the computational costs of the general PAK-UCB algorithm.

### 6.2. Experiment Settings

**Setup 1. Prompt-based selection among standard text-to-image models.** The first set of experiments are on the setup illustrated in Figure 1(a), where we generate images from Stable Diffusion v1.5 and PixArt-$\alpha$. The results show that PAK-UCB-poly3 outperforms the baseline algorithm and attains a high optimal-pick-ratio, which shows that it can identify the optimal model conditioned to the prompt (see Figure 3). Additionally, we provide numerical results on various T2I generative models, including UniDiffuser and DeepFloyd (see Figure 4). Prompts are uniformly randomly selected from the MS-COCO dataset under two categories: 'dog'/'car' (Figure 3), 'train'/'baseball-bat' (Figure 4(a)),

'elephant'/'fire-hydrant' (Figure 4(b)), and 'carrot'/'bowl' (Figure 4(c)). The input of the PAK-UCB, Naive-KRR, and RFF-UCB-RBF methods is the embedded prompt that is output by the pretrained CLIP-ViT-B-32-laion2B-e16 model from the open_clip repository[7]. For LinUCB and KernelUCB baselines, we also concatenate the one-hot encoded vector of the model index to the CLIP-embedded prompt. Full results can be found in Figure 8 in the Appendix.

**Setup 2. Prompt-based selection of LLMs.** In these experiments, we focus on prompt-based selection of large language models (LLMs) on various tasks, including Sudoku-solving and code generation (code completion and translation). In the first setup, we select LLMs to solve code generation tasks. The model is given a Python code completion problem (sampled from the first 164 tasks in Big-CodeBench (Zhuo et al., 2024)) or C++-to-Java code translation problem (sampled from the HumanEval-X benchmark (Zheng et al., 2023)). Sample prompts can be found in Figure 18 and 19 in the Appendix. Particularly, the Gemini-2.5-Flash-preview model attains a higher pass@1 rate on code completion (55.91% versus 13.84%) while underperforms for C++-to-Java translation (24.39% versus 40.91%) compared to the Qwen-Plus model. The rewards are binary, indicating whether the generated code can pass all the test cases. The input of the PAK-UCB, Naive-KRR, and RFF-UCB-RBF methods is the 768-dimensional embedded prompt that is output by the RoBERTa model (for code completion) or the CodeBERT model[8] (for code translation). For LinUCB and KernelUCB baselines, we also concatenate the one-hot encoded vector of the model index to the embedded prompt. The results (Figures 6(a) and 20) show that the proposed RFF-UCB-RBF method queries from the better model conditioned to the task category (Figure 5).

---

[7] https://github.com/mlfoundations/open_clip/tree/main

[8] https://huggingface.co/microsoft/codebert-base

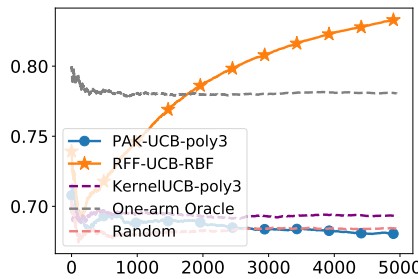

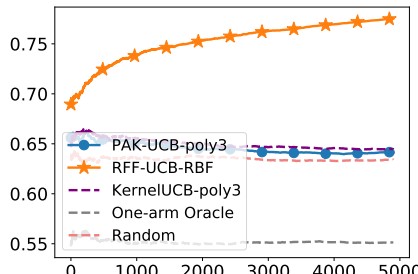

(a) Code generation: Claude-3.5-Haiku and Gemini-2.5-Flash-preview

(b) Sudoku-solving: Deepseek-Chat and o3-mini

*Figure 5.* Prompt-based selection of LLMs (Setup 2): Optimal-pick-ratio (OPR) is reported.

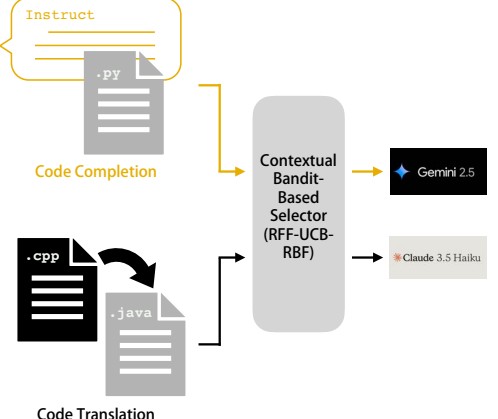

*Figure 6.* Prompt-based selection of LLMs on code generation task (Setup 2): The Gemini-2.5-Flash-preview model attains higher pass rate on the code completion task while underperforms on the code translation task. Our proposed RFF-UCB-RBF algorithm queries solutions from the better model conditioned to the task category.

In the second setup, we utilize two LLMs, including o3-mini and Deepseek-Chat, to solve $9 \times 9$ Sudoku problems, which contains 5-30 blank entries. Sample prompt can be found in Figure 16 in the Appendix. The o3-mini model attains a higher success rate on harder Sudoku problems with more than 15 blank entries (70.82% versus 50.42%), while slightly underperforms on easier Sudoku problems (86.85% versus 93.6%). The rewards are binary, indicating the solution is correct or wrong. The input of the PAK-UCB, Naive-KRR, and RFF-UCB-RBF methods is the 768-dimensional embedded prompt that is output by the RoBERTa model[9]. For LinUCB and KernelUCB baselines, we also concatenate the one-hot encoded vector of the model index to the RoBERTa-embedded prompt. The results are summarized in Figures 5(b) and 17, which show that the proposed RFF-

UCB-RBF algorithm outperforms all the baseline methods.

**Setup 3. Adaptation to new prompt types and models.**
We consider scenarios where new text-to-image models or prompt types are introduced after the initial deployment. At the beginning of the first experiment, Stable Diffusion v1.5 and PixArt-$\alpha$ are available, and UniDiffuser is introduced after 2500 iterations. Prompts are uniformly randomly selected from the MS-COCO dataset under categories 'train' and 'baseball-bat'. In the second experiment, we generate samples from both PixArt-$\alpha$ and UniDiffuser. In the first 1k iterations, the prompts are uniformly randomly selected from a pool that initially includes categories 'person' and 'bicycle' in the MS-COCO dataset. Then, categories 'airplane', 'bus', 'train', and 'truck' are added to the pool after each 1k iterations. The results show that PAK-UCB-poly3 can well adapt to new prompt types and generators (see Figure 14). The input context follows Setup 1.

**Setup 4. Synthetic experiments on other conditional generation tasks.** We provide numerical results on image-captioning (image-to-text) and text-to-video (T2V) task under synthetic setups in Appendix C.3.

## 7. Conclusion

In this work, we investigated prompt-based selection of generative models using a contextual bandit algorithm, which can identify the best available generative model for a given text prompt. We propose two kernel-based algorithms, including PAK-UCB and RFF-UCB, to perform this selection task. Our numerical results on LLMs (text-to-text), text-to-image, text-to-video, and image-captioning tasks demonstrate the effectiveness of the proposed framework in scenarios where the available generative models have varying performance rankings depending on the type of prompt. A potential future direction is to extend the online selection to capture the diversity and novelty of generated data. Also, developing cost-aware online selection methods, similar to the cost-aware CB algorithm in (Hu et al., 2025b), will be a relevant direction for future studies.

[9] https://huggingface.co/docs/transformers/en/model_doc/roberta

## Acknowledgements

The work of Farzan Farnia is partially supported by a grant from the Research Grants Council of the Hong Kong Special Administrative Region, China, Project 14209920, and is partially supported by CUHK Direct Research Grants with CUHK Project No. 4055164 and 4937054. Also, the authors would like to thank the anonymous reviewers for their constructive feedback and suggestions.

## Impact Statement

This paper presents work whose goal is to advance the field of Machine Learning. There are many potential societal consequences of our work, none which we feel must be specifically highlighted here.

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

# A. Proof in Section 5.1

The technical challenge in analyzing PAK-UCB is that predictions in later iterations make use of previous outcomes. Hence, the rewards $\{s_i\}_{i \in \Psi_g}$ are not independent if the index set $\Phi_g$ is updated each time when model $g$ is chosen (Line 6 of Algorithm 2). To address this problem, we leverage a standard approach used in prior works (Auer, 2003; Chu et al., 2011; Valko et al., 2013) and present a variant of PAK-UCB in Algorithm 3, which is called Sup-PAK-UCB.

---

**Algorithm 3** Sup-PAK-UCB

---

**Require:** total iterations $T \in \mathbb{N}_+$, set of generators $\mathcal{G} = [G]$, prompt distribution $\rho \in \Delta(\mathcal{Y})$, score function $s : \mathcal{Y} \times \mathcal{X} \to [-1, 1]$, positive definite kernel $k : \mathcal{Y} \times \mathcal{Y} \to \mathbb{R}$, regularization and exploration parameters $\alpha, \eta \geq 0$, function COMPUTE_UCB in Algorithm 2

**Initialize:** observation sequence $\mathcal{D} \leftarrow \varnothing$ and index sets $\{\Psi_g^m \leftarrow \varnothing\}_{m=1}^M$ for all $g \in \mathcal{G}$, where $M \leftarrow \log T$

1: **for** iteration $t = 1, 2, \cdots, T$ **do**
2:     Prompt $y_t \sim \rho$ is revealed.
3:     Set stage $m \leftarrow 1$ and $\widehat{\mathcal{G}}^1 \leftarrow \mathcal{G}$.
4:     **repeat**
5:         Compute $\{(\widehat{\mu}_g^m, \widehat{\sigma}_g^m) \leftarrow \text{COMPUTE\_UCB}(\mathcal{D}, y_t, \Psi_g^m)\}_{g \in \widehat{\mathcal{G}}^m}$.
6:         Set $\widehat{s}_g^m(y_t) \leftarrow \widehat{\mu}_g^m + \eta \widehat{\sigma}_g^m$ and $\widetilde{s}_g^m(y_t) \leftarrow \widehat{\mu}_g^m + (2\eta + \sqrt{\alpha})\widehat{\sigma}_g^m$ for all $g \in \widehat{\mathcal{G}}^m$.
7:         **if** $(\eta + \sqrt{\alpha})\widehat{\sigma}_g^m \leq 1/\sqrt{T}$ for all $g \in \widehat{\mathcal{G}}^m$ **then**
8:             Pick model $g_t \leftarrow \arg\max \widehat{s}_g^m(y)$.
9:         **else if** $(2\eta + \sqrt{\alpha})\widehat{\sigma}_g^m \leq 2^{1-m}$ for all $g \in \widehat{\mathcal{G}}^m$ **then**
10:            $\widehat{\mathcal{G}}^{m+1} \leftarrow \left\{g \in \widehat{\mathcal{G}}^m : \widetilde{s}_g^m(y_t) \geq \max_{g \in \widehat{\mathcal{G}}^m} \widetilde{s}_g^m(y_t) - 2^{2-m}\right\}$.
11:            Set stage $m \leftarrow m + 1$.
12:         **else**
13:            Pick $g_t \in \widehat{\mathcal{G}}^m$ such that $(2\eta + \sqrt{\alpha})\widehat{\sigma}_g^m > 2^{1-m}$.
14:            Update $\Psi_{g_t}^m \leftarrow \Psi_{g_t}^m \cup \{t\}$.
15:         **end if**
16:     **until** a model $g_t$ is selected
17:     Sample an answer $x_t \sim P_{g_t}(\cdot|y_t)$ and compute the score $s_t \leftarrow s(y_t, x_t)$.
18:     Update $\mathcal{D} \leftarrow \mathcal{D} \cup \{(y_t, s_t)\}$.
19: **end for**

---

## A.1. Regret Analysis of Sup-PAK-UCB

**Theorem 3** (Regret of Sup-PAK-UCB). *Under Assumption 1, with probability at least $1 - \delta$, the regret of* Sup-PAK-UCB *with $\eta = \sqrt{2\log(2GT/\delta)}$ is bounded by*

$$\text{Regret}(T) \leq \widetilde{O}\left((1 + \sqrt{\alpha})\sqrt{d_{\text{eff}}\left(1 + \frac{1}{\alpha}\right)GT}\right), \tag{4}$$

*where $d_{\text{eff}}$ is a data-dependent quantity defined in Lemma 5 and logarithmic factors are hidden in the notation $\widetilde{O}(\cdot)$.*

**Notations.** To facilitate the analysis, we add an subscript $t$ to all the notations in Algorithm 3 to indicate they are quantities computed at the $t$-th iteration, i.e., $\widehat{\mu}_{g,t}^m$ and $\widehat{\sigma}_{g,t}^m$ (first appeared Line 5),$\widehat{s}_{g,t}^m$ and $\widetilde{s}_{g,t}^m$ (first appeared Line 6), $\widehat{\mathcal{G}}_t^m$, and $\Psi_{g,t}^m$. In addition, we define $g_{\star,t} := \arg\max_{g \in \mathcal{G}} s_g(y_t)$ as the optimal model for prompt $y_t$ and $\widehat{g}_{\star,t}^m := \arg\max_{g \in \widehat{\mathcal{G}}_t^m} \widehat{s}_{g,t}^m$ is optimistic model at stage $m$.[10]

*Proof of Theorem 3.* Let $\mathcal{T}_1 := \cup_{m \in [M], g \in \mathcal{G}} \Psi_{g,T+1}^m$ and $\mathcal{T}_0 := [T] \setminus \mathcal{T}_1$. Note that $\mathcal{T}_0$ and $\mathcal{T}_1$ are sets of iterations such that the model is selected in Lines 8 and 13 of Algorithm 3, respectively.

---

[10]Note that Line 14 of Algorithm 3 is rewritten as "$\Psi_{g_t,t+1}^m \leftarrow \Psi_{g_t,t+1}^m \cup \{t\}$, $\Psi_{g_t,t+1}^{m'} \leftarrow \Psi_{g_t,t}^{m'}$ for any $m' \neq m$, and $\Psi_{g,t+1}^{m'} \leftarrow \Psi_{g,t}^{m'}$ for any $g \neq g_t$ and $m \in [M]$". In addition, we set $\Psi_{g,t+1}^m \leftarrow \Psi_{g,t+1}^m$ for all $g \in \mathcal{G}$ and $m \in [M]$ in Line 8.

**1. Regret incurred in $\mathcal{T}_0$.** For any $t \in [T]$, let $m_t$ denote the stage that model $g_t$ is picked at the $t$-th iteration. We have that

$$
\begin{aligned}
\sum_{t \in \mathcal{T}_0} (s_\star(y_t) - s_{g_t}(y_t)) &= \sum_{t \in \mathcal{T}_0} (s_\star(y_t) - \widehat{s}_{g_\star,t,t}^{m_t}(y) + \underbrace{\widehat{s}_{g_\star,t,t}^{m_t}(y) - \widehat{s}_{g_t}^{m_t}(y_t)}_{\leq 0 \text{ by Line 8}} + \widehat{s}_{g_t}^{m_t}(y_t) - s_{g_t}(y_t)) \\
&\leq (\eta + \sqrt{\alpha}) \sum_{t \in \mathcal{T}_0} \widehat{\sigma}_{g_\star,t,t}^{m_t} + (3\eta + \sqrt{\alpha}) \sum_{t \in \mathcal{T}_0} \widehat{\sigma}_{g_t,t}^{m_t} \\
&\leq O(\sqrt{T}),
\end{aligned}
\tag{5}
$$

where the first inequality holds by the fact that $g_{\star,t} \in \widehat{\mathcal{G}}_t^{m_t}$.

**2. Regret incurred in $\mathcal{T}_1$.** Note that

$$
\begin{aligned}
\sum_{t \in \mathcal{T}_1} (s_\star(y_t) - s_{g_t}(y_t)) &= \sum_{g \in \mathcal{G}} \sum_{m \in [M]} \sum_{t \in \Psi_{g,T+1}^m} (s_\star(y_t) - s_{g_t}(y_t)) \\
&\leq \sum_{g \in \mathcal{G}} \sum_{m \in [M]} 2^{3-m} \cdot |\Psi_{g,T+1}^m|,
\end{aligned}
\tag{6}
$$

where the inequality holds by the last statement in Lemma 3. It remains to bound $|\Psi_{g,T+1}^m|$. First note that for any $m \in [M]$, we have that

$$
\sum_{t \in \Psi_{g,T+1}^m} (2\eta + \sqrt{\alpha}) \widehat{\sigma}_{g,t}^m > 2^{1-m} \cdot |\Psi_{g,T+1}^m|
$$

from Line 13 of Algorithm 3. In addition, by a similar statement of (Valko et al., 2013, Lemma 4), which is stated in Lemma 5, we have that

$$
\sum_{t \in \Psi_{g,T+1}^m} \widehat{\sigma}_{g,t}^m \leq \widetilde{O}\left( \sqrt{d_{\text{eff}}\left(1 + \frac{1}{\alpha}\right) |\Psi_{g,T+1}^m|} \right),
\tag{7}
$$

where $d_{\text{eff}}$ is defined therein and logarithmic factors are hidden in the notation $\widetilde{O}(\cdot)$. Plugging in Equation (6) results in

$$
\begin{aligned}
\sum_{t \in \mathcal{T}_1} (s_\star(y_t) - s_{g_t}(y_t)) &\leq \widetilde{O}\left( \sum_{g \in \mathcal{G}} \sum_{m \in [M]} \sqrt{d_{\text{eff}}\left(1 + \frac{1}{\alpha}\right) |\Psi_{g,T+1}^m|} \right) \\
&\leq \widetilde{O}\left( \sqrt{GM} \sqrt{d_{\text{eff}}\left(1 + \frac{1}{\alpha}\right) \sum_{g \in \mathcal{G}} \sum_{m \in [M]} |\Psi_{g,T+1}^m|} \right) \\
&\leq \widetilde{O}\left( \sqrt{d_{\text{eff}}\left(1 + \frac{1}{\alpha}\right) GT} \right),
\end{aligned}
\tag{8}
$$

where the second inequality holds by Cauchy-Schwarz inequality.

**3. Putting everything together.** Combining Inequalities (5) and (8) leads to

$$
\text{Regret}(T) = \left( \sum_{t \in \mathcal{T}_0} + \sum_{t \in \mathcal{T}_1} \right) (s_\star(y_t) - s_{g_t}(y_t)) \leq \widetilde{O}\left( (1 + \sqrt{\alpha}) \sqrt{d_{\text{eff}}\left(1 + \frac{1}{\alpha}\right) GT} \right),
$$

which concludes the proof. $\square$

### A.2. Auxiliary Lemmas

**Lemma 2** (Auer (2003), Lemma 14). *For any iteration $t \in [T]$, model $g \in \mathcal{G}$, and stage $m \in [M]$, the set of rewards $\{s_i\}_{i \in \Psi_{g,t}^m}$ are independent random variables such that $\mathbb{E}[s_i] = s_g(y_i)$.*

**Lemma 3.** *With probability at least $1 - (MGT)\delta$, for any iteration $t \in [T]$ and stage $m \in [M]$ in Algorithm 3, the following statements hold:*

- $|\widehat{\mu}_{g,t}^m - s_g(y_t)| \leq (2\eta + \sqrt{\alpha})\widehat{\sigma}_{g,t}^m$ *for any* $g \in \widehat{\mathcal{G}}_t^m$,

- $\arg\max_{g \in \mathcal{G}} s_g(y_t) \in \widehat{\mathcal{G}}_t^m$, *and*

- $s_\star(y_t) - s_g(y_t) \leq 2^{3-m}$ *for any* $g \in \widehat{\mathcal{G}}_t^m$.

*Proof.* The first statement holds by both Lemma 2 and Lemma 4, and a uniform bound over all $t \in [T], g \in \mathcal{G}$, and $m \in [M]$.

To show the second statement, first note that $g_{\star,t} \in \widehat{\mathcal{G}}_t^1$. Assume $g_{\star,t} \in \widehat{\mathcal{G}}_t^m$ for some $m \in [M-1]$. Let $\widetilde{s}_{g,t}^m := \widehat{\mu}_{g,t}^m + (2\eta + \sqrt{\alpha})\widehat{\sigma}_{g,t}^m$ (computed in Line 10 in the algorithm). Then, by the first statement, we obtain that

$$
\widetilde{s}_{g_{\star,t},t}^m - \max_{g \in \widehat{\mathcal{G}}_t^m} \widetilde{s}_{g,t}^m = \underbrace{\widetilde{s}_{g_{\star,t},t}^m - s_\star(y_t)}_{\geq 0} + s_\star(y_t) - \max_{g \in \widehat{\mathcal{G}}_t^m} \left\{ \widetilde{s}_{g,t}^m - s_g(y_t) + s_g(y_t) \right\}
$$

$$
\geq - \max_{g \in \widehat{\mathcal{G}}_t^m} \left\{ \widetilde{s}_{g,t}^m - s_g(y_t) \right\}
$$

$$
\geq 2(2\eta + \sqrt{\alpha})\widehat{\sigma}_{g,t}^m \geq 2 \cdot (-2^{1-m}) = -2^{2-m},
$$

where the last inequality holds by Line 9 of the algorithm.

To show the last statement, note that for any $g \in \widehat{\mathcal{G}}_t^m$, it holds that

$$
s_\star(y_t) - s_g(y_t) = \underbrace{s_\star(y_t) - \widetilde{s}_{g_{\star,t},t}^m}_{\leq 0} + \widetilde{s}_{g_{\star,t},t}^m - \left( s_g(y_t) - \widetilde{s}_{g,t}^m + \widetilde{s}_{g,t}^m \right)
$$

$$
\leq 2(2\eta + \sqrt{\alpha})\widehat{\sigma}_g^{m,t} + |\widetilde{s}_{g_{\star,t},t}^m - \widetilde{s}_{g,t}^m|
$$

$$
\leq 2 \cdot 2^{1-m} + 2^{2-m} \leq 2^{3-m},
$$

which concludes the proof. $\square$

**Lemma 4** (Optimism). *Let* $\Psi_g \subseteq [T]$ *be an index set such that the set of scores* $\{s_t : t \in \Psi_g\}$ *are independent random variables. Then, under Assumption 1, with probability at least* $1 - \delta$, *the quantity* $\widehat{\mu}_g$ *computed in function* COMPUTE_UCB$(\mathcal{D}, y, \Psi_g)$ *satisfies that*

$$
|\widehat{\mu}_g - s_g(y)| \leq (2\eta + \sqrt{\alpha})\widehat{\sigma}_g, \tag{9}
$$

*where* $\eta = \sqrt{2\log(2/\delta)}$. *Hence, it holds that* $\widehat{s}_g = \widehat{\mu}_g + (2\eta + \sqrt{\alpha})\widehat{\sigma}_g \geq s_g(y)$.

*Proof.* We rewrite the proof using the notations in Section 5. Obviously, Equation (9) holds when the index set $\Psi_g$ is empty. In the following, we consider non-empty $\Psi_g$. Let $\Phi_g := [\phi(y_i)^\top]_{i \in \Psi_g}$. Note that $k_y = [k(y, y_i)]_{i \in \Psi_g}^\top = \Phi_g(\phi(y))$ and $K = [k(y_i, y_j)]_{i,j \in \Psi_g} = \Phi_g \Phi_g^\top$. We have

$$
\begin{aligned}
\widehat{\mu}_g - s_g(y) &= (\phi(y))^\top \Phi_g^\top (K + \alpha I)^{-1} v - (\phi(y))^\top w_g^\star \\
&= (\phi(y))^\top (\Phi_g^\top \Phi_g + \alpha I)^{-1} \Phi_g^\top v - (\phi(y))^\top (\Phi_g^\top \Phi_g + \alpha I)^{-1} (\Phi_g^\top \Phi_g + \alpha I) w_g^\star \\
&= \underbrace{(\phi(y))^\top (\Phi_g^\top \Phi_g + \alpha I)^{-1} \Phi_g^\top (v - \Phi_g w_g^\star)}_{(a)} - \underbrace{\alpha (\phi(y))^\top (\Phi_g^\top \Phi_g + \alpha I)^{-1} w_g^\star}_{(b)},
\end{aligned} \tag{10}
$$

where the second equation holds by the positive definiteness of both matrices $(K + \alpha I)$ and $(\Phi_g^\top \Phi_g + \alpha I)$ and hence

$$
\Phi_g^\top (K + \alpha I)^{-1} = (\Phi_g^\top \Phi_g + \alpha I)^{-1} \Phi_g^\top. \tag{11}
$$

**1. Bounding Term (a).** Note that the scores $\{s_t : t \in \Psi_g\}$ are independent by the construction of $\Phi_g$ and $\mathbb{E}[s_t] = (w_g^\star)^\top \phi(y_t)$, we have that

$$
(\phi(y))^\top (\Phi_g^\top \Phi_g + \alpha I)^{-1} \Phi_g^\top (v - \Phi_g w_g^\star) = \sum_{i=1}^{|\Psi_g|} [(\phi(y))^\top (\Phi_g^\top \Phi_g + \alpha I)^{-1} \Phi_g^\top]_i \cdot [v - \Phi_g w_g^\star]_i
$$

are the summation of zero mean independent random variables, where we denote by $[\cdot]_i$ the $i$-th element of a vector. Further, each variable satisfies that

$$
\begin{aligned}
&\left|[(\phi(y))^\top (\Phi_g^\top \Phi_g + \alpha I)^{-1} \Phi_g^\top]_i \cdot [v - \Phi_g w_g^\star]_i\right| \\
&\leq \|(\phi(y))^\top (\Phi_g^\top \Phi_g + \alpha I)^{-1} \Phi_g^\top\| \cdot |[v - \Phi_g w_g^\star]_i| \\
&\leq \sqrt{(\phi(y))^\top (\Phi_g^\top \Phi_g + \alpha I)^{-1} \Phi_g^\top \Phi_g (\Phi_g^\top \Phi_g + \alpha I)^{-1} (\phi(y))} \cdot (1 + \|w_g^\star\|) \\
&\leq 2\widehat{\sigma}_g,
\end{aligned}
$$

where the last inequality holds by $\|w_g^\star\| \leq 1$ and the second inequality holds by

$$
\begin{aligned}
\widehat{\sigma}_g &= \alpha^{-\frac{1}{2}} \sqrt{k(y,y) - k_y^\top (K + \alpha I)^{-1} k_y} \\
&= \alpha^{-\frac{1}{2}} \sqrt{(\phi(y))^\top (\phi(y)) - (\phi(y))^\top \Phi_g^\top (K + \alpha I)^{-1} \Phi_g (\phi(y))} \\
&= \alpha^{-\frac{1}{2}} \sqrt{(\phi(y))^\top \left(I - (\Phi_g^\top \Phi_g + \alpha I)^{-1} \Phi_g^\top \Phi_g\right) (\phi(y))} \\
&= \sqrt{(\phi(y))^\top (\Phi_g^\top \Phi_g + \alpha I)^{-1} (\phi(y))},
\end{aligned}
$$

Then, by Azuma-Hoeffding inequality, it holds that

$$
\begin{aligned}
&\mathbb{P}\left(|(\phi(y))^\top (\Phi_g^\top \Phi_g + \alpha I)^{-1} \Phi_g^\top (v - \Phi_g w_g^\star)| > 2\eta\widehat{\sigma}_g\right) \\
&\leq 2\exp\left(-\frac{\widehat{\sigma}_g^2 \eta^2}{2|\Psi_g|\widehat{\sigma}_g^2}\right) \\
&\leq 2\exp(-\eta^2/2).
\end{aligned} \tag{12}
$$

Solving the above inequality to be smaller than $\delta$ leads to $\eta = \sqrt{2\log(2/\delta)}$.

**2. Bounding Term (b).** By the Cauchy-Schwarz inequality, it holds that

$$
\begin{aligned}
&\left|(\phi(y))^\top (\Phi_g^\top \Phi_g + \alpha I)^{-1} w_g^\star\right| \\
&\leq \|w_g^\star\| \cdot \|(\phi(y))^\top (\Phi_g^\top \Phi_g + \alpha I)^{-1}\| \\
&= \|w_g^\star\| \cdot \sqrt{(\phi(y))^\top (\Phi_g^\top \Phi_g + \alpha I)^{-1} \alpha^{-1} \alpha I (\Phi_g^\top \Phi_g + \alpha I)^{-1} (\phi(y))} \\
&\leq \alpha^{-1/2} \sqrt{(\phi(y))^\top (\Phi_g^\top \Phi_g + \alpha I)^{-1} (\Phi_g^\top \Phi_g + \alpha I)(\Phi_g^\top \Phi_g + \alpha I)^{-1} (\phi(y))} \\
&= \alpha^{-1/2} \widehat{\sigma}_g,
\end{aligned} \tag{13}
$$

where the second inequality holds by the positive definiteness of $\Phi_g^\top \Phi_g$.

**3. Putting everything together.** Plugging (12) and (13) in (10) and setting $\delta = 2\exp(-\eta^2/2)$ concludes the proof. $\quad\square$

**Lemma 5** (Valko et al. (2013), Lemma 4). *For any model $g \in \mathcal{G}$ and stage $m \in [M]$, let $\lambda_{g,1}^m \geq \lambda_{g,2}^m \geq \cdots$ denote the eigenvalues (in the decreasing order) of the matrix $(\Phi_g^m)^\top \Phi_g^m + \alpha I$, where $\Phi_g^m = [\phi(y_i)^\top]_{i \in \Psi_{g,T+1}^m}$. Then, for any iteration $t \in [T]$, it holds that*

$$
\sum_{t \in \Psi_{g,T+1}^m} \widehat{\sigma}_{g,t}^m \leq \widetilde{O}\left(\sqrt{d_{\text{eff}}\left(1 + \frac{1}{\alpha}\right)|\Psi_{g,T+1}^m|}\right),
$$

*where $d_{\text{eff}} := \max_{g \in \mathcal{G}, m \in [M]} \min\{j \in \mathbb{N}_+ : j\alpha \log T \geq \Lambda_{g,j}^m\}$ and $\Lambda_{g,j}^m := \sum_{i > j} \lambda_{g,i}^m - \alpha$ is the effective dimension.*

# B. Missing Algorithms and Proofs in Section 5.2

## B.1. The RFF-UCB Algorithm

---

**Algorithm 4** RFF-UCB: PAK-UCB with RFF implementation

---

**Require:** iteration $T$, generators $\mathcal{G} = [G]$, prompt distribution $\rho$, score function $s : \mathcal{Y} \times \mathcal{X} \to [-1, 1]$, positive definite kernel $k$, regularization and exploration parameters $\alpha, \eta \geq 0$, function COMPUTE_UCB_RFF (Algorithm 5)
**Initialize:** observation sequence $\mathcal{D} \leftarrow \varnothing$ and index set $\Psi_g \leftarrow \varnothing$ for all $g \in \mathcal{G}$
1: **for** iteration $t = 1, 2, \cdots, T$ **do**
2:     Prompt $y_t \sim \rho$ is revealed.
3:     Compute $(\widetilde{\mu}_g, \widetilde{\sigma}_g) \leftarrow$ COMPUTE_UCB_RFF$(\mathcal{D}, y_t, \Psi_g)$ and set $\widehat{s}_g \leftarrow \widetilde{\mu}_g + \eta \widetilde{\sigma}_g$ for each $g \in \mathcal{G}$.
4:     Pick model $g_t \leftarrow \arg \max_{g \in \mathcal{G}} \{\widehat{s}_g\}$.
5:     Sample $x_t \sim P_{g_t}(\cdot | y_t)$ and receive score $s_t$.
6:     Update $\mathcal{D} \leftarrow \mathcal{D} \cup \{(y_t, s_t)\}$ and $\Psi_{g_t} \leftarrow \Psi_{g_t} \cup \{t\}$.
7: **end for**

---

**Algorithm 5** Compute UCB with Random Fourier Features

---

**Require:** the Fourier transform $p$ of a positive definite shift-invariant kernel $k(y, y') = k(y - y')$, regularization and exploration parameters $\alpha, \eta \geq 0$
**Initialize:** number of features $D$
1: **function** COMPUTE_UCB_RFF$(\mathcal{D}, y, \Psi_g)$
2:     **if** $\Psi_g$ is empty **then**
3:         $\widetilde{\mu}_g \leftarrow +\infty, \widetilde{\sigma}_g \leftarrow +\infty$.
4:     **else**
5:         Draw $\omega_1, \cdots, \omega_D \overset{\text{i.i.d.}}{\sim} p$
6:         Define mapping $\varphi(y') \leftarrow \sqrt{\frac{1}{D}} \cdot [\cos(w_1^\top y), \sin(w_1^\top y), \cdots, \cos(w_D^\top y), \sin(w_D^\top y)]^\top$ for any $y' \in \mathbb{R}^d$.
7:         Set $\widetilde{\Phi}_g \leftarrow [\varphi(y_i)^\top]_{i \in \Psi_g}$ and $v \leftarrow [s_i]_{i \in \Psi_g}^\top$.
8:         $\widetilde{\mu}_g \leftarrow (\varphi(y))^\top (\widetilde{\Phi}_g^\top \widetilde{\Phi}_g + \alpha I_{2D})^{-1} \widetilde{\Phi}_g^\top v$.
9:         $\widetilde{\sigma}_g \leftarrow \alpha^{-\frac{1}{2}} (1 - (\varphi(y))^\top (\widetilde{\Phi}_g^\top \widetilde{\Phi}_g + \alpha I_{2D})^{-1} \widetilde{\Phi}_g^\top \widetilde{\Phi}_g (\varphi(y)))^{\frac{1}{2}}$.
10:     **end if**
11:     **return** $(\widetilde{\mu}_g, \widetilde{\sigma}_g)$.
12: **end function**

---

**Analysis of Lemma 1.** Solving KRR with $n$ regression data requires $\Theta(n^3)$ time and $\Theta(n^2)$ space. Hence, by the convexity of the cubic and quadratic functions, the time for COMPUTE_UCB scales with $\Theta(\sum_{g \in \mathcal{G}} n_g^3) = O(t^3/G^2)$, and the space scales with $\Theta(\sum_{g \in \mathcal{G}} n_g^2) = O(t^2/G)$, where $n_g := |\Psi_g|$ is the visitation to any model $g \in \mathcal{G}$ up to iteration $t$, and we have $\sum_{g \in \mathcal{G}} n_g = t$. On the other hand, solving KRR with $n$ regression data and random features of size $s$ requires $O(nD^2)$ time and $O(nD)$ space. Therefore, the time for COMPUTE_UCB_RFF scales with $O(\sum_{g \in \mathcal{G}} n_g D^2) = O(tD^2)$, and the space scales with $O(\sum_{g \in \mathcal{G}} n_g D) = O(tD)$.

## B.2. Regret analysis of Sup-RFF-UCB

**Algorithm description.** We present Sup-RFF-UCB in Algorithm 6, where we utilize function COMPUTE_UCB_RFF to compute the UCB values. To achieve the regret bound (4), an important problem is to design (adaptive) error thresholds, i.e., $\epsilon_{\text{RFF}}$ and $\Delta_{\text{RFF}}$, when computing UCB at each stage $m$ and iteration $t$. We prove the regret bound in the following theorem.

---

**Algorithm 6** Sup-RFF-UCB

---

**Require:** total iterations $T \in \mathbb{N}_+$, set of generators $\mathcal{G} = [G]$, prompt distribution $\rho \in \Delta(\mathcal{Y})$, score function $s : \mathcal{Y} \times \mathcal{X} \to [-1,1]$, positive definite kernel $k : \mathcal{Y} \times \mathcal{Y} \to \mathbb{R}$, regularization and exploration parameters $\alpha, \eta \geq 0$, function COMPUTE_UCB_RFF in Algorithm 5, parameters $D_{g,t}^m$, $\mathcal{B}_{g,1,t}^m$, and $\mathcal{B}_{g,2,t}^m$

**Initialize:** observation sequence $\mathcal{D} \leftarrow \varnothing$ and index sets $\{\Psi_g^m \leftarrow \varnothing\}_{m=1}^M$ for all $g \in \mathcal{G}$, where $M \leftarrow \log T$

1: **for** iteration $t = 1, 2, \cdots, T$ **do**
2:      Prompt $y_t \sim \rho$ is revealed.
3:      Set stage $m \leftarrow 1$ and $\widehat{\mathcal{G}}^1 \leftarrow \mathcal{G}$.
4:      **repeat**
5:          Compute $\{(\widetilde{\mu}_g^m, \widetilde{\sigma}_g^m) \leftarrow \text{COMPUTE\_UCB\_RFF}(\mathcal{D}, y_t, \Psi_g^m)\}_{g \in \widehat{\mathcal{G}}^m}$ with RFF dimension $D_{g,t}^m$
6:          Set $\widehat{s}_g^m := \widetilde{\mu}_g^m + \eta \widetilde{\sigma}_g^m$ for all $g \in \widehat{\mathcal{G}}^m$.
7:          Set $\widetilde{s}_g^m \leftarrow \widetilde{\mu}_g^m + \mathcal{B}_{g,1,t}^m + (2\eta + \sqrt{\alpha})(\widetilde{\sigma}_g^m + \mathcal{B}_{g,2,t}^m)$ for all $g \in \widehat{\mathcal{G}}^m$.
8:          **if** $(\eta + \sqrt{\alpha})\widetilde{\sigma}_g^m \leq 1/\sqrt{T}$ for all $g \in \widehat{\mathcal{G}}^m$ **then**
9:             Pick model $g_t \leftarrow \arg\max_{g \in \widehat{\mathcal{G}}^m} \widehat{s}_g^m$.
10:        **else if** $\mathcal{B}_{g,1,t}^m + (2\eta + \sqrt{\alpha})(\widetilde{\sigma}_g^m + \mathcal{B}_{g,2,t}^m) \leq 2^{1-m}$ for all $g \in \widehat{\mathcal{G}}^m$ **then**
11:             $\widehat{\mathcal{G}}^{m+1} \leftarrow \left\{ g \in \widehat{\mathcal{G}}^m : \widetilde{s}_g^m \geq \max_{g \in \widehat{\mathcal{G}}^m}\{\widetilde{s}_g^m - 2^{2-m}\} \right\}$.
12:             Set stage $m \leftarrow m + 1$.
13:        **else**
14:             Pick $g_t \in \widehat{\mathcal{G}}^m$ such that $\mathcal{B}_{g,1,t}^m + (2\eta + \sqrt{\alpha})(\widetilde{\sigma}_g^m + \mathcal{B}_{g,2,t}^m) > 2^{1-m}$.
15:             Update $\Psi_{g_t}^m \leftarrow \Psi_{g_t}^m \cup \{t\}$.
16:        **end if**
17:      **until** a model $g_t$ is selected
18:      Sample an answer $x_t \sim P_{g_t}(\cdot|y_t)$ and compute the score $s_t \leftarrow s(y_t, x_t)$.
19:      Update $\mathcal{D} \leftarrow \mathcal{D} \cup \{(y_t, s_t)\}$.
20: **end for**

---

**Theorem 4** (Regret of Sup-RFF-UCB). *With probability at least $1 - \delta$, the regret of running Sup-RFF-UCB with exploration parameter $\eta = \sqrt{2 \log(4GT/\delta)}$ and regularization $\alpha \leq 1$ for $T$ iterations is bounded by (4). Specifically, at stage $m \in [M]$ at iteration $t \in [T]$, the parameters satisfy*

$$D_{g,t}^m \geq \max \left\{ \frac{4(d+2)\zeta_{\epsilon_{\text{RFF}_{g,t}^m}}}{(\epsilon_{\text{RFF}_{g,t}^m})^2} \left[ \frac{2}{1 + \frac{2}{d}} \log \frac{\sigma_p}{\epsilon_{\text{RFF}_{g,t}^m}} + \log \frac{\beta_d}{\delta} \right], \frac{8|\Psi_g|}{3\alpha}(\Delta_{\text{RFF}_{g,t}^m})^{-2} \log \left( \frac{32 \iota_\alpha(K)}{\delta} \right) \right\}, \tag{14}$$

$$\mathcal{B}_{g,1,t}^m = |\Psi_{g,t}^m| \epsilon_{\text{RFF}_{g,t}^m} + \alpha^{-1}|\Psi_{g,t}^m| \Delta_{\text{RFF}_{g,t}^m}(|\Psi_{g,t}^m| + \alpha), \tag{15}$$

$$\mathcal{B}_{g,2,t}^m = (\alpha t)^{-\frac{1}{2}} + t^{\frac{3}{2}} \alpha^{-\frac{1}{2}} \left( \alpha^{-1} \Delta_{\text{RFF}_{g,t}^m}(|\Psi_{g,t}^m| + \alpha) + 2 \epsilon_{\text{RFF}_{g,t}^m} \right), \tag{16}$$

*where $\iota_\alpha$ is defined in Lemma 10, $\zeta$ and $\sigma_p$ are defined in Lemma 11, and the error thresholds are given by*

$$\epsilon_{\text{RFF}_{g,t}^m} \leq t^{-2}, \quad \Delta_{\text{RFF}_{g,t}^m} \leq \alpha t^{-2}(|\Psi_{g,t}^m| + \alpha)^{-1}. \tag{17}$$

*Proof.* Note that $\mathcal{B}_{g,1,t}^m \leq O(t^{-1})$ and $\mathcal{B}_{g,2,t}^m \leq O(t^{-\frac{1}{2}})$. The proof is based on Lemma 6. For iterations in $\mathcal{T}_0$ (model $g_t$ is picked in Line 8 of Algorithm 6), we have

$$\sum_{t \in \mathcal{T}_0} (s_\star(y_t) - s_{g_t}(y_t))$$

$$= \sum_{t \in \mathcal{T}_0} (s_\star(y_t) - \widehat{s}_{g_\star,t,t}^{m_t}(y) + \underbrace{\widehat{s}_{g_\star,t,t}^{m_t}(y) - \widehat{s}_{g_t,t}^{m_t}(y_t)}_{\leq 0 \text{ by Line 8}} + \widehat{s}_{g_t,t}^{m_t}(y_t) - s_{g_t}(y_t))$$

$$\leq \sum_{t \in \mathcal{T}_0} (\underbrace{s_\star(y_t) - \widetilde{s}_{g_\star,t,t}^{m_t}(y)}_{\leq 0 \text{ by Lemma 6}} + \widetilde{s}_{g_\star,t,t}^{m_t}(y) - \widehat{s}_{g_\star,t,t}^{m_t}(y) + \underbrace{\widehat{s}_{g_t,t}^{m_t}(y_t) - \widetilde{s}_{g_t}^{m_t}(y_t)}_{\leq 0 \text{ by definitions}} + \widetilde{s}_{g_t}^{m_t}(y_t) - s_{g_t}(y_t))$$

$$= \sum_{t \in \mathcal{T}_0} \left( \mathcal{B}^{m_t}_{g_\star,t,1} + (\eta + \sqrt{\alpha})\widetilde{\sigma}^m_{g_\star,t} + 2\mathcal{B}^{m_t}_{g_t,t,1} + 2(2\eta + \sqrt{\alpha})\widetilde{\sigma}^{m_t}_{g_t,t} + (2\eta + \sqrt{\alpha})(\mathcal{B}^{m_t}_{g_\star,t,2} + 2\mathcal{B}^{m_t}_{g_t,t,2}) \right)$$

$$\leq 4 \sum_{t \in \mathcal{T}_0} \underbrace{(\eta + \sqrt{\alpha})(\widetilde{\sigma}^{m_t}_{g_\star,t} + \widetilde{\sigma}^{m_t}_{g_t,t})}_{\leq 2/\sqrt{T} \text{ by Line 8}} + \sum_{t \in \mathcal{T}_0} \left( \mathcal{B}^{m_t}_{g_\star,t,1} + \mathcal{B}^{m_t}_{g_t,t,1} + 2(2\eta + \sqrt{\alpha})(\mathcal{B}^{m_t}_{g_\star,t,2} + \mathcal{B}^{m_t}_{g_t,t,2}) \right)$$

$$\leq \widetilde{O} \left( \sqrt{T} + \sum_{t \in \mathcal{T}_0} \left( t^{-1} + (1 + \sqrt{\alpha})\alpha^{-\frac{1}{2}} t^{-\frac{1}{2}} \right) \right)$$

$$\leq \widetilde{O} \left( (1 + \alpha^{-\frac{1}{2}})\sqrt{T} \right),$$

where the last inequality holds by the fact that each $t \in [T]$ appears in at most one index set. Further, for iterations in $\mathcal{T}_1$ (model $g_t$ is picked in Line 13 of Algorithm 3), the third statement in Lemma 6 ensures that

$$\sum_{t \in \mathcal{T}_1} (s_\star(y_t) - s_{g_t}(y_t))$$

$$\leq \sum_{g \in \mathcal{G}} \sum_{m \in [M]} 2^{3-m} \cdot |\Psi^m_{g,T+1}|$$

$$< 4 \sum_{g \in \mathcal{G}} \sum_{m \in [M]} \sum_{t \in \Psi^m_{g,T+1}} \left( \mathcal{B}^{m_t}_{g_t,1,t} + (2\eta + \sqrt{\alpha}) \left( \widetilde{\sigma}^m_{g,t} + \mathcal{B}^{m_t}_{g_t,2,t} \right) \right)$$

$$\leq 4 \sum_{g \in \mathcal{G}} \sum_{m \in [M]} \sum_{t \in \Psi^m_{g,T+1}} \left( \mathcal{B}^{m_t}_{g_t,1,t} + (2\eta + \sqrt{\alpha}) \left( \widehat{\sigma}^m_{g,t} + 2\mathcal{B}^{m_t}_{g_t,2,t} \right) \right)$$

$$= 4 \sum_{g \in \mathcal{G}} \sum_{m \in [M]} \left( \sum_{t \in \Psi^m_{g,T+1}} \left( \mathcal{B}^{m_t}_{g_t,1,t} + 2(2\eta + \sqrt{\alpha})\mathcal{B}^{m_t}_{g_t,2,t} \right) + (2\eta + \sqrt{\alpha}) \sum_{t \in \Psi^m_{g,T+1}} \widehat{\sigma}^m_{g,t} \right)$$

$$= 4 \sum_{g \in \mathcal{G}} \sum_{m \in [M]} \left( \sum_{t \in \Psi^m_{g,T+1}} \left( \mathcal{B}^{m_t}_{g_t,1,t} + 2(2\eta + \sqrt{\alpha})\mathcal{B}^{m_t}_{g_t,2,t} \right) \right) + 4 \sum_{g \in \mathcal{G}} \sum_{m \in [M]} \left( (2\eta + \sqrt{\alpha}) \sum_{t \in \Psi^m_{g,T+1}} \widehat{\sigma}^m_{g,t} \right).$$

Note that the upper bound of the second term has been derived in Equation (7). It remains to bound the first term. Essentially, we will find a sequence of error thresholds, and hence the number of features defined in Inequality (20), such that the first term is bounded by $\widetilde{O}((1 + \alpha^{-\frac{1}{2}})\sqrt{T})$.

For any iteration $t \in [T]$, model $g \in \mathcal{G}$, and stage $m \in [M]$, we define $K^m_{g,t} := \Phi^m_{g,t}(\Phi^m_{g,t})^\top$, where $\Phi^m_{g,t} := [\phi(y_i)^\top]_{i \in \Psi^m_{g,t}}$. First, by the definition of $\mathcal{B}_{g,1}$ in Equation (15), we have

$$\sum_{g \in \mathcal{G}, m \in [M]} \sum_{t \in \Psi^m_{g,T+1}} \mathcal{B}^m_{g_t,1,t}$$

$$= \sum_{g \in \mathcal{G}, m \in [M]} \sum_{t \in \Psi^m_{g,T+1}} \left( |\Psi^m_{g,t}| \epsilon_{\mathrm{RFF}}{}^m_{g,t} + \alpha^{-1}|\Psi^m_{g,t}| \Delta_{\mathrm{RFF}}{}^m_{g,t}(|\Psi^m_{g,t}| + \alpha) \right) \qquad (18)$$

$$\leq \sum_{t=1}^{T} (t^{-1} + t^{-1})$$

$$\leq O(\log T),$$

where the first inequality holds by the fact that each $t \in [T]$ appears in at most one index set. On the other hand, by the

definition of $\mathcal{B}_{g,2}$ in Equation (16), we derive

$$
\begin{aligned}
&\sum_{g \in \mathcal{G}, m \in [M]} \sum_{t \in \Psi_{g,T+1}^m} \mathcal{B}_{g_t,2,t}^m \\
&\leq \sum_{g \in \mathcal{G}, m \in [M]} \sum_{t \in \Psi_{g,T+1}^m} \left( (\alpha t)^{-\frac{1}{2}} + t^{\frac{3}{2}} \alpha^{-\frac{1}{2}} \left( \alpha^{-1} \Delta_{\mathrm{RFF}_{g,t}^m}(|\Psi_{g,t}^m| + \alpha) + 2\,\epsilon_{\mathrm{RFF}_{g,t}^m} \right) \right) \\
&\leq \sum_{t=1}^{T} \left( 4\alpha^{-\frac{1}{2}} t^{-\frac{1}{2}} \right) \\
&\leq \widetilde{O}\left( \alpha^{-\frac{1}{2}} \sqrt{T} \right).
\end{aligned}
\tag{19}
$$

Therefore, we conclude the proof. □

## B.3. Auxiliary Lemmas

**Lemma 6.** *With probability at least $1 - (2MGT)\delta$, for any iteration $t \in [T]$ and stage $m \in [M]$, the following hold:*

- $|\widetilde{\mu}_{g,t}^m - s_g(y_t)| \leq \mathcal{B}_{g,1,t}^m + (2\eta + \sqrt{\alpha})(\widetilde{\sigma}_{g,t}^m + \mathcal{B}_{g,2,t}^m)$ *for any $g \in \widehat{\mathcal{G}}_t^m$,*

- $\arg\max_{g \in \mathcal{G}} s_g(y_t) \in \widehat{\mathcal{G}}_t^m$, *and*

- $s_\star(y_t) - s_g(y_t) \leq 2^{3-m}$ *for any $g \in \widehat{\mathcal{G}}_t^m$.*

*where the first statement is guaranteed by Theorem 7, and $\mathcal{B}_{g,1,t}^m$ and $\mathcal{B}_{g,2,t}^m$ are given in (15) and (16), respectively.*

*Proof.* The proof follows the exact same analysis of Lemma 3. □

**Lemma 7** (Optimistic KRR estimators with RFF). *Assume the error thresholds input to Algorithm 5 satisfy that $\Delta_{\mathrm{RFF}}, \epsilon_{\mathrm{RFF}} \leq 1/2$. Let regularization $\alpha \leq 1$. With probability at least $1 - 2\delta$, the quantity $\widetilde{\mu}_g$ computed by function $\mathrm{COMPUTE\_UCB\_RFF}$ satisfies that*

$$
|\widetilde{\mu}_g - s_g(y)| \leq \mathcal{B}_{g,1} + (2\eta + \sqrt{\alpha})(\widetilde{\sigma}_g + \mathcal{B}_{g,2}),
$$

*with the number of features satisfying Inequality (20) and bonus terms $\mathcal{B}_{g,1}$ and $\mathcal{B}_{g,2}$ given by Equations (15) and (16), where $\eta = \sqrt{2\log(2/\delta)}$. Hence, it holds that $\widetilde{s}_g = \widetilde{\mu}_g + \mathcal{B}_{g,1} + (2\eta + \sqrt{\alpha})(\widetilde{\sigma}_g + \mathcal{B}_{g,2}) \geq s_g(y)$.*

*Proof.* The proof is based on the following two lemmas, which analyze the concentration error of the quantities $\widetilde{\mu}_g$ and $\widetilde{\sigma}_g$. Finally, combining Lemmas 4, 8, and 9, we derive that

$$
\begin{aligned}
|\widetilde{\mu}_g - s_g(y)| &\leq |\widetilde{\mu}_g - \widehat{\mu}_g| + |\widehat{\mu}_g - s_g(y)| \\
&\leq \mathcal{B}_{g,1} + (2\eta + \sqrt{\alpha})\widehat{\sigma}_g \\
&\leq \mathcal{B}_{g,1} + (2\eta + \sqrt{\alpha})(\widetilde{\sigma}_g + \mathcal{B}_{g,2}),
\end{aligned}
$$

which concludes the proof. □

**Lemma 8** (Concentration of mean using RFF). *Let $\Delta_{\mathrm{RFF}}, \epsilon_{\mathrm{RFF}} \leq 1/2$ and the regularization parameter $\alpha \leq 1$. Let $\Psi_g \subseteq [T]$ be an index set such that the set of scores $\{s_t : t \in \Psi_g\}$ are independent random variables. Then, with probability at least $1 - \delta$, the quantity $\widetilde{\mu}_g$ computed by function $\mathrm{COMPUTE\_UCB\_RFF}$ satisfies that*

$$
|\widetilde{\mu}_g - \widehat{\mu}_g| \leq |\Psi_g|\,\epsilon_{\mathrm{RFF}} + \alpha^{-1}|\Psi_g|\,\Delta_{\mathrm{RFF}}(|\Psi_g| + \alpha)
$$

*with the number of features*

$$
D \geq \max\left\{ \frac{4(d+2)\zeta_{\epsilon_{\mathrm{RFF}}}}{\epsilon_{\mathrm{RFF}}^2} \left[ \frac{2}{1 + \frac{2}{d}} \log\frac{\sigma_p}{\epsilon_{\mathrm{RFF}}} + \log\frac{\beta_d}{\delta} \right], \frac{8|\Psi_g|}{3\alpha} \Delta_{\mathrm{RFF}}^{-2} \log\left( \frac{32\iota_\alpha(K)}{\delta} \right) \right\},
\tag{20}
$$

*where $\widehat{\mu}_g = (\phi(y))^\top \Phi_g^\top (K + \alpha I)^{-1} v$, and $\sigma_p^2, \zeta_{\epsilon_{\mathrm{RFF}}}, \beta_d$ and $\iota_\alpha(\cdot)$ are quantities defined in Lemmas 10 and 11, respectively.*

*Proof.* For convenience, we define $\widetilde{k}_y := \widetilde{\Phi}_g(\varphi(y)) \in \mathbb{R}^{|\Psi_g|}$, $Q := (K + \alpha I)^{-1} \in \mathbb{R}^{|\Psi_g| \times |\Psi_g|}$, and $\widetilde{Q} := (\widetilde{K} + \alpha I)^{-1} \in \mathbb{R}^{|\Psi_g| \times |\Psi_g|}$, where $\widetilde{K} := \widetilde{\Phi}_g \widetilde{\Phi}_g^\top$. Using the same notations in the proof of Lemma 4, we obtain that

$$
\begin{aligned}
|\widetilde{\mu}_g - \widehat{\mu}_g| &= \left| (\varphi(y))^\top \widetilde{\Phi}_g^\top (\widetilde{K} + \alpha I)^{-1} v - k_y^\top (K + \alpha I)^{-1} v \right| \\
&= \left| \widetilde{k}_y^\top \widetilde{Q} v - k_y^\top Q v \right| \\
&\leq \left| \widetilde{k}_y^\top (\widetilde{Q} - Q) v \right| + \left| (\widetilde{k}_y - k_y)^\top Q v \right|,
\end{aligned}
\tag{21}
$$

where we use Equation (11) to derive $\widetilde{\mu}_g = (\varphi(y))^\top (\widetilde{\Phi}_g^\top \widetilde{\Phi}_g + \alpha I)^{-1} \widetilde{\Phi}_g^\top v = (\varphi(y))^\top \widetilde{\Phi}_g^\top (\widetilde{K} + \alpha I)^{-1} v$ in the first equation.

**1. Bounding $|(\widetilde{k}_y - k_y)^\top Q v|$.** Note that $\mathcal{Y} \subset \mathbb{S}^{d-1}$. We evoke (Sutherland & Schneider, 2015, Proposition 1), which is rewritten in Lemma 11 using our notations. For a desired threshold $\epsilon_{\text{RFF}} > 0$, set

$$
D \geq \frac{4(d+2)\zeta_{\epsilon_{\text{RFF}}}}{\epsilon_{\text{RFF}}^2} \left[ \frac{2}{1 + \frac{2}{d}} \log \frac{\sigma_p}{\epsilon_{\text{RFF}}} + \log \frac{\beta_d}{\delta} \right].
$$

Then, with probability at least $1 - \frac{\delta}{2}$, it holds that $\sup_{y,y' \in \mathcal{Y}} |(\varphi(y))^\top \varphi(y') - k(y, y')| \leq \epsilon_{\text{RFF}}$, and hence $\|\widetilde{k}_y - k_y\|_\infty \leq \epsilon_{\text{RFF}}$. Therefore, we obtain

$$
|(\widetilde{k}_y - k_y)^\top Q v| \leq \|\widetilde{k}_y - k_y\|_2 \cdot \|Q\|_2 \cdot \|v\|_2 \leq \epsilon_{\text{RFF}} \sqrt{|\Psi_g|} \cdot (1 + \alpha)^{-1} \cdot \sqrt{|\Psi_g|} \leq |\Psi_g| \epsilon_{\text{RFF}},
\tag{22}
$$

where the second inequality holds by $\|Q\|_2 = \lambda_{\min}^{-1}(K + \alpha I) \leq (1 + \alpha)^{-1}$ and $\|v\|_\infty \leq 1$.

**2. Bounding $|\widetilde{k}_y^\top (\widetilde{Q} - Q) v|$.** Note that

$$
|\widetilde{k}_y^\top (\widetilde{Q} - Q) v| \leq \|\widetilde{k}_y\|_2 \cdot \|\widetilde{Q} - Q\|_2 \cdot \|v\|_2 \leq \sqrt{|\Psi_g|} \cdot \|\widetilde{Q} - Q\|_2 \cdot \sqrt{|\Psi_g|},
$$

where the first inequality holds by the fact that $(\varphi(y))^\top \varphi(y_i) \leq 1$. To bound $\|\widetilde{Q} - Q\|_2$, we evoke (Avron et al., 2017, Theorem 7), which is rewritten in Lemma 10. For a desired threshold $\Delta_{\text{RFF}} \leq 1/2$, the following inequality holds with probability at least $1 - \frac{\delta}{2}$:

$$
(1 - \Delta_{\text{RFF}})(K + \alpha I) \preceq \widetilde{K} + \alpha I
$$

for $D \geq \frac{8|\Psi_g|}{3\alpha} \Delta_{\text{RFF}}^{-2} \log(32 \iota_\alpha(K)/\delta)$. By Sherman-Morrison-Woodbury formula, i.e., $A^{-1} - B^{-1} = A^{-1}(B - A)B^{-1}$ where $A$ and $B$ are invertible, we derive

$$
\begin{aligned}
\|\widetilde{Q} - Q\|_2 &= \|(\widetilde{K} + \alpha I)^{-1} - (K + \alpha I)^{-1}\|_2 \\
&\leq \|(\widetilde{K} + \alpha I)^{-1}\|_2 \cdot \|(K + \alpha I) - (\widetilde{K} + \alpha I)\|_2 \cdot \|(K + \alpha I)^{-1}\|_2 \\
&\leq \alpha^{-1} \Delta_{\text{RFF}}(\|K\|_2 + \alpha) \leq \alpha^{-1} \Delta_{\text{RFF}}(|\Psi_g| + \alpha),
\end{aligned}
\tag{23}
$$

where the second inequality holds by the fact that $\|\widetilde{Q}\|_2 = \|(\widetilde{K} + \alpha I)^{-1}\|_2 \leq \alpha^{-1}$, $\|Q\|_2 = \|(K + \alpha I)^{-1}\|_2 \leq (1 + \alpha)^{-1}$ and $\|(K + \alpha I) - (\widetilde{K} + \alpha I)\|_2 \leq \|\Delta_{\text{RFF}}(K + \alpha I)\|_2 \leq \Delta_{\text{RFF}}(\|K\|_2 + \alpha)$, and the last inequality holds by the fact that $\|K\|_2 \leq |\Psi_g|$ for any shift-invariant and PSD kernel.

**3. Putting everything together.** Combining Equations (22) and (23), with probability at least $1 - \delta$, it holds that

$$
|\widetilde{\mu}_g - \widehat{\mu}_g| \leq |\Psi_g| \epsilon_{\text{RFF}} + \alpha^{-1} |\Psi_g| \Delta_{\text{RFF}}(|\Psi_g| + \alpha)
$$

when

$$
D \geq \max \left\{ \frac{4(d+2)\zeta_{\epsilon_{\text{RFF}}}}{\epsilon_{\text{RFF}}^2} \left[ \frac{2}{1 + \frac{2}{d}} \log \frac{\sigma_p}{\epsilon_{\text{RFF}}} + \log \frac{\beta_d}{\delta} \right], \frac{8|\Psi_g|}{3\alpha} \Delta_{\text{RFF}}^{-2} \log \left( \frac{32 \iota_\alpha(K)}{\delta} \right) \right\},
$$

which concludes the proof. $\quad\square$

**Lemma 9** (Concentration of variance using RFF). *Conditioned on the successful events in Lemma 8, the quantity $\widetilde{\sigma}_g$ computed by function* COMPUTE_UCB_RFF *satisfies that*

$$|\widetilde{\sigma}_g - \widehat{\sigma}_g| \leq \max\{(\alpha t)^{-\frac{1}{2}}, t^{\frac{3}{2}}\alpha^{-\frac{1}{2}}\left(\alpha^{-1}\Delta_{\text{RFF}}(|\Psi_g| + \alpha) + 2\,\epsilon_{\text{RFF}}\right)\}, \tag{24}$$

*where $\widehat{\sigma}_g = \alpha^{-\frac{1}{2}}\sqrt{k(y,y) - k_y^\top(K + \alpha I)^{-1}k_y}$.*

*Proof.* We use the same notations in the proof of Lemma 8. Note that if $\widetilde{\sigma}_g, \widehat{\sigma}_g \leq (\alpha t)^{-\frac{1}{2}}$, we have $|\widetilde{\sigma}_g - \widehat{\sigma}_g| \leq (\alpha t)^{-\frac{1}{2}}$. On the other hand, if one of $\widetilde{\sigma}_g, \widehat{\sigma}_g \geq (\alpha t)^{-\frac{1}{2}}$, we have

$$
\begin{aligned}
&|\widetilde{\sigma}_g - \widehat{\sigma}_g| \\
=&\alpha^{-\frac{1}{2}}\left|\sqrt{1 - (\varphi(y))^\top\widetilde{\Phi}_g^\top(\widetilde{K} + \alpha I)^{-1}\widetilde{\Phi}_g(\varphi(y))} - \sqrt{1 - k_y^\top(K + \alpha I)^{-1}k_y}\right| \\
\leq&\sqrt{t}\alpha^{-\frac{1}{2}}\left|(\varphi(y))^\top\widetilde{\Phi}_g^\top(\widetilde{K} + \alpha I)^{-1}\widetilde{\Phi}_g(\varphi(y)) - k_y^\top(K + \alpha I)^{-1}k_y\right| \\
=&\sqrt{t}\alpha^{-\frac{1}{2}}\left|\widetilde{k}_y^\top\widetilde{Q}\widetilde{k}_y - k_y^\top Q k_y\right| \\
\leq&\sqrt{t}\alpha^{-\frac{1}{2}}\left(\left|\widetilde{k}_y^\top(\widetilde{Q} - Q)\widetilde{k}_y\right| + \left|(\widetilde{k}_y - k_y)^\top Q\widetilde{k}_y\right| + \left|k_y^\top Q(\widetilde{k}_y - k_y)\right|\right) \\
\leq&\sqrt{t}\alpha^{-\frac{1}{2}}\left(\|\widetilde{k}_y\|_2^2\|\widetilde{Q} - Q\|_2 + \|\widetilde{k}_y - k_y\|_2\|\widetilde{k}_y\|_2\|Q\|_2 + \|\widetilde{k}_y - k_y\|_2\|k_y\|_2\|Q\|_2\right) \\
\leq&t^{\frac{3}{2}}\alpha^{-\frac{1}{2}}\left(\alpha^{-1}\Delta_{\text{RFF}}(|\Psi_g| + \alpha) + 2\,\epsilon_{\text{RFF}}\right),
\end{aligned}
\tag{25}
$$

where we utilize Inequality (23) to obtain the last inequality. We conclude the proof. $\square$

**Lemma 10** (Avron et al. (2017), Theorem 7). *Let $K = [k(y_i, y_j)]_{i,j\in[n]}$ denote the Gram matrix of $\{y_i \in \mathbb{R}^d\}_{i=1}^n$, where $k$ is a shift-invariant kernel function. Let $\Delta \leq 1/2$ and $\delta \in (0,1)$. Assume that $\|K\|_2 \geq \alpha$. If we use $D \geq \frac{8n}{3\alpha}\Delta^{-2}\log(16\iota_\alpha(K)/\delta)$ random Fourier features, then with probability at least $1 - \delta$, it holds that*

$$(1 - \Delta)(K + \alpha I) \preceq \widetilde{K} + \alpha I \preceq (1 + \Delta)(K + \alpha I),$$

*where $\iota_\alpha(K) := \text{Tr}[(K + \alpha I)^{-1}K]$ and we denote by $\widetilde{K} = [(\varphi(y_i))^\top(\varphi(y_j))]_{i,j\in[n]}$ the approximated Gram matrix using $s \in \mathbb{N}_+$ random Fourier features, where $\varphi : \mathbb{R}^d \to \mathbb{R}^s$ is the feature mapping.*

**Lemma 11** (Adapted from (Sutherland & Schneider, 2015, Proposition 1)). *Let $k$ be a continuous shift-invariant positive-definite function $k(y, y') = k(y - y')$ defined on $\mathcal{Y} \subseteq \mathbb{R}^d$, with $k(0) = 1$ and such that $\nabla^2 k(0)$ exists. Suppose $\mathcal{Y}$ is compact, with diameter $\text{diam}(\mathcal{Y})$. Let $\varphi(y)$ be as in Equation (1). For any $\epsilon > 0$, let*

$$\zeta_\epsilon := \min\left(1, \sup_{y,y'\in\mathcal{Y}} \frac{1}{2} + \frac{1}{2}k(2y, 2y') - k(y, y')^2 + \frac{1}{3}\epsilon\right),$$

$$\beta_d := \left(\left(\frac{d}{2}\right)^{-\frac{d}{d+2}} + \left(\frac{d}{2}\right)^{\frac{2}{d+2}}\right)2^{\frac{6d+2}{d+2}}.$$

*Then, assuming only for the second statement that $\epsilon \leq \sigma_p\text{diam}(\mathcal{Y})$,*

$$\mathbb{P}\left(\sup_{y,y'\in\mathcal{Y}}|(\varphi(y))^\top\varphi(y) - k(y, y')| \geq \epsilon\right) \leq 66\left(\frac{\sigma_p\text{diam}(\mathcal{Y})}{\epsilon}\right)^2\exp\left(-\frac{D\epsilon^2}{4(d+2)}\right),$$

*where $\sigma_p^2 := \mathbb{E}_p[\|\omega\|^2]$ is the second moment of the Fourier transform of $k$.[11] Thus, we can achieve an embedding with pointwise error no more than $\epsilon$ with probability at least $1 - \delta$ as long as*

$$D \geq \frac{4(d+2)\zeta_\epsilon}{\epsilon^2}\left[\frac{2}{1 + \frac{2}{d}}\log\frac{\sigma_p\text{diam}(\mathcal{Y})}{\epsilon} + \log\frac{\beta_d}{\delta}\right].$$

---

[11] For the RBF kernel with parameter $\sigma^2$, i.e., $k_{\text{RBF}}^\sigma(y, y') = \exp(-\frac{1}{2\sigma^2}\|y - y'\|_2^2)$, we have $\sigma_{p_{\text{RBF}}}^2 = \frac{d}{\sigma^2}$.

# C. Additional Experimental Details and Results

## C.1. Results on Setups 1 and 2

**1. Varying rankings of text-to-image models.** We provide more examples showing that prompt-based generative models can outperform for text prompts from certain categories while underperforming for other text categories (see Figures 7, 9, and 11).

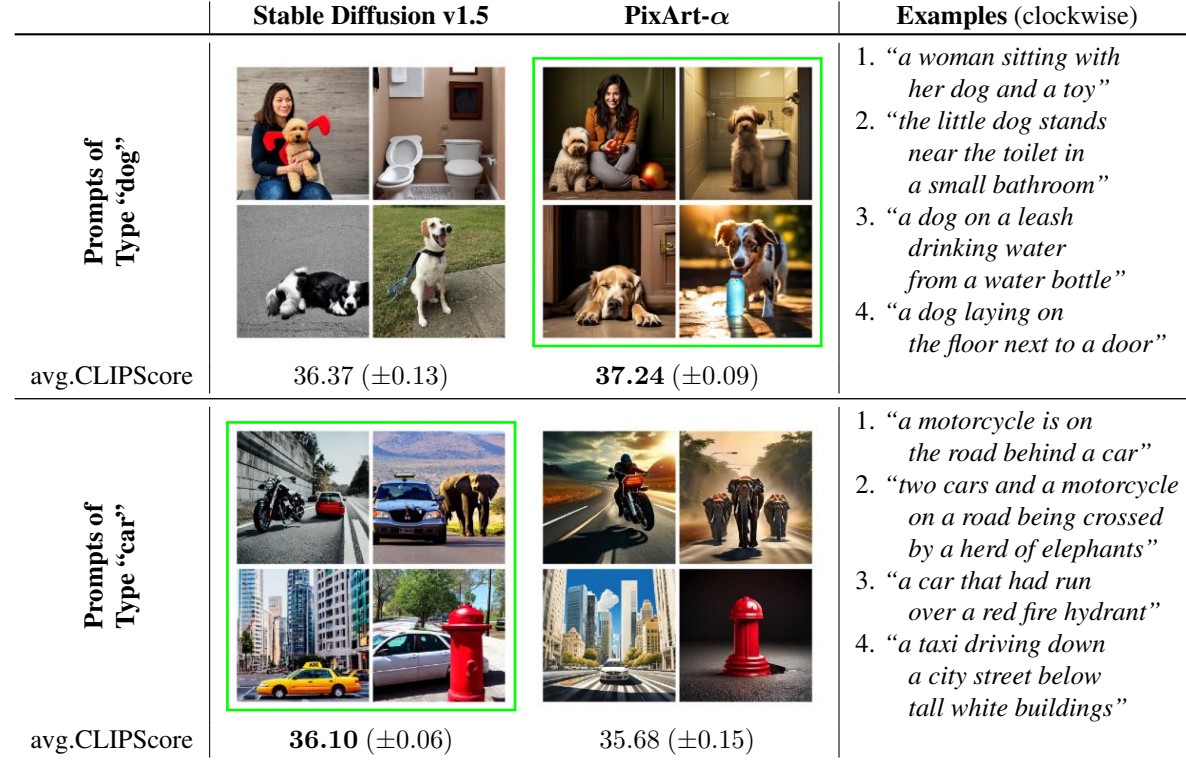

| | **Stable Diffusion v1.5** | **PixArt-$\alpha$** | **Examples** (clockwise) |
|---|---|---|---|
| Prompts of Type "dog" | | | 1. *"a woman sitting with her dog and a toy"* 
 2. *"the little dog stands near the toilet in a small bathroom"* 
 3. *"a dog on a leash drinking water from a water bottle"* 
 4. *"a dog laying on the floor next to a door"* |
| avg.CLIPScore | 36.37 ($\pm$0.13) | **37.24** ($\pm$0.09) | |
| Prompts of Type "car" | | | 1. *"a motorcycle is on the road behind a car"* 
 2. *"two cars and a motorcycle on a road being crossed by a herd of elephants"* 
 3. *"a car that had run over a red fire hydrant"* 
 4. *"a taxi driving down a city street below tall white buildings"* |
| avg.CLIPScore | **36.10** ($\pm$0.06) | 35.68 ($\pm$0.15) | |

*Figure 7.* Prompt-based generated images from Stable Diffusion v1.5 and PixArt-$\alpha$: Stable Diffusion v1.5 attains a higher CLIPScore in generating type-2 prompts (36.10 versus 35.68) while underperforms for type-1 prompts (36.37 versus 37.24).

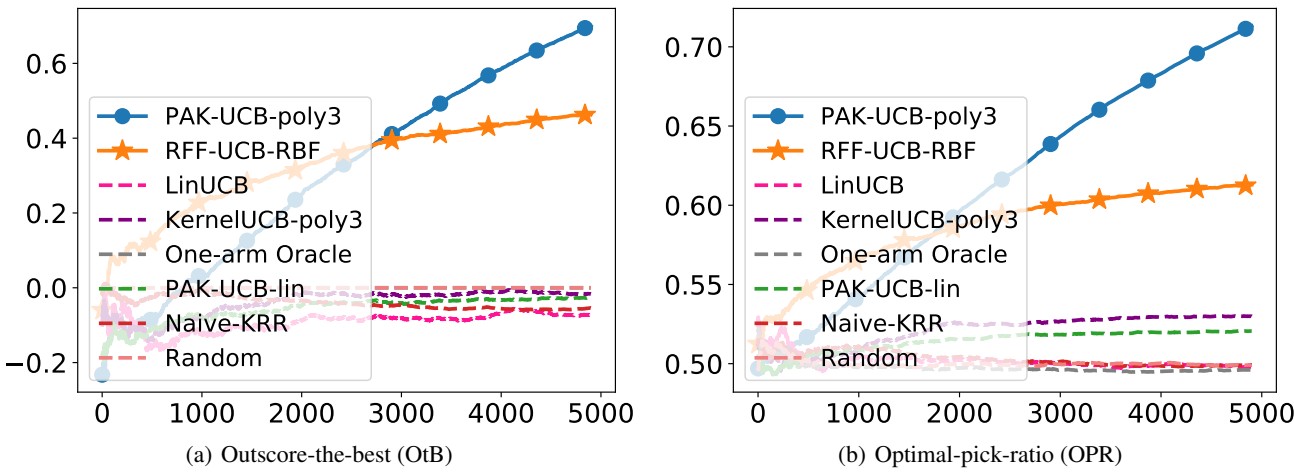

(a) Outscore-the-best (OtB)

(b) Optimal-pick-ratio (OPR)

*Figure 8.* Prompt-based selection between Stable Diffusion v1.5 and PixArt-$\alpha$ (see Figure 7): Results are averaged over 20 trials.

**2. Prompt-based selection over three T2I models.** See Figure 13.

|  | **UniDiffuser** | **PixArt-$\alpha$** | **Examples** (clockwise) |
|---|---|---|---|
| Prompts of Type "train" | 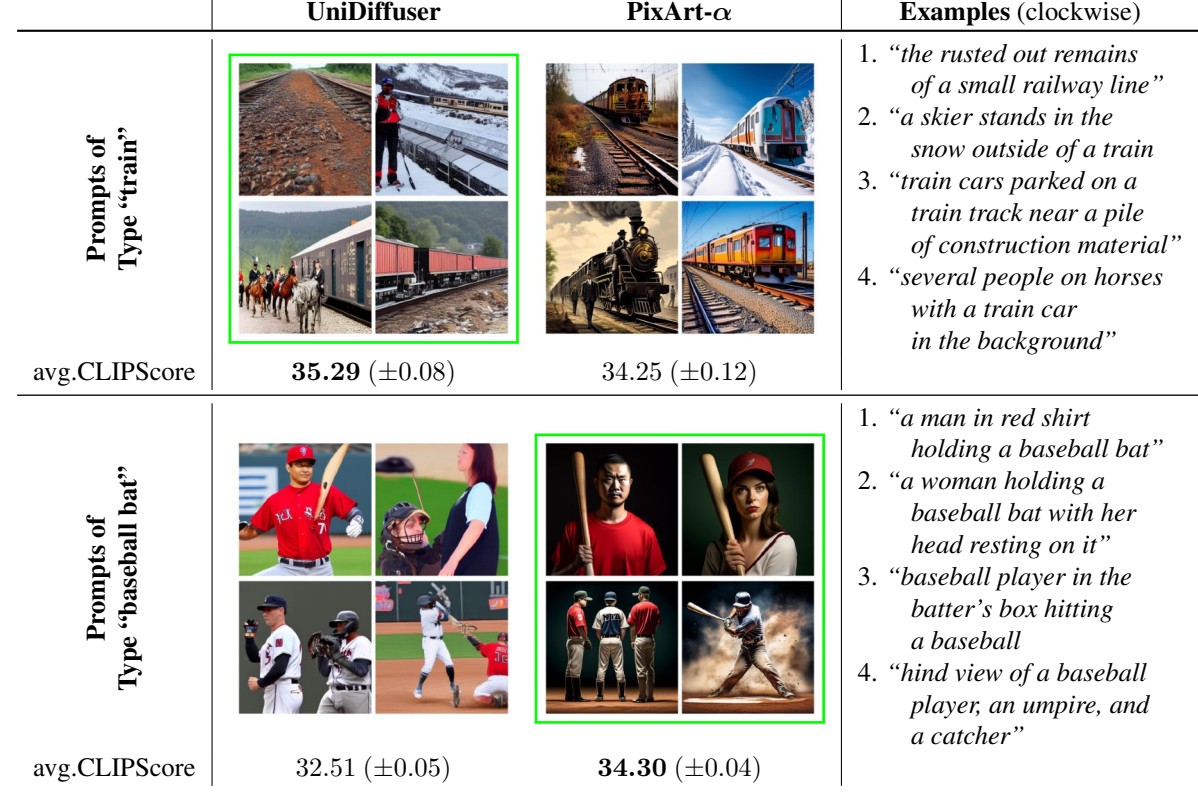 | | 1. *"the rusted out remains of a small railway line"* 
 2. *"a skier stands in the snow outside of a train* 
 3. *"train cars parked on a train track near a pile of construction material"* 
 4. *"several people on horses with a train car in the background"* |
| avg.CLIPScore | **35.29** ($\pm$0.08) | 34.25 ($\pm$0.12) | |
| Prompts of Type "baseball bat" | | | 1. *"a man in red shirt holding a baseball bat"* 
 2. *"a woman holding a baseball bat with her head resting on it"* 
 3. *"baseball player in the batter's box hitting a baseball* 
 4. *"hind view of a baseball player, an umpire, and a catcher"* |
| avg.CLIPScore | 32.51 ($\pm$0.05) | **34.30** ($\pm$0.04) | |

*Figure 9.* Prompt-based generated images from UniDiffuser (Bao et al., 2023a) and PixArt-$\alpha$: UniDiffuser attains a higher CLIPScore in generating type "train" prompts (35.29 versus 34.25) while underperforms for type "baseball bat" prompts (32.51 versus 34.30).

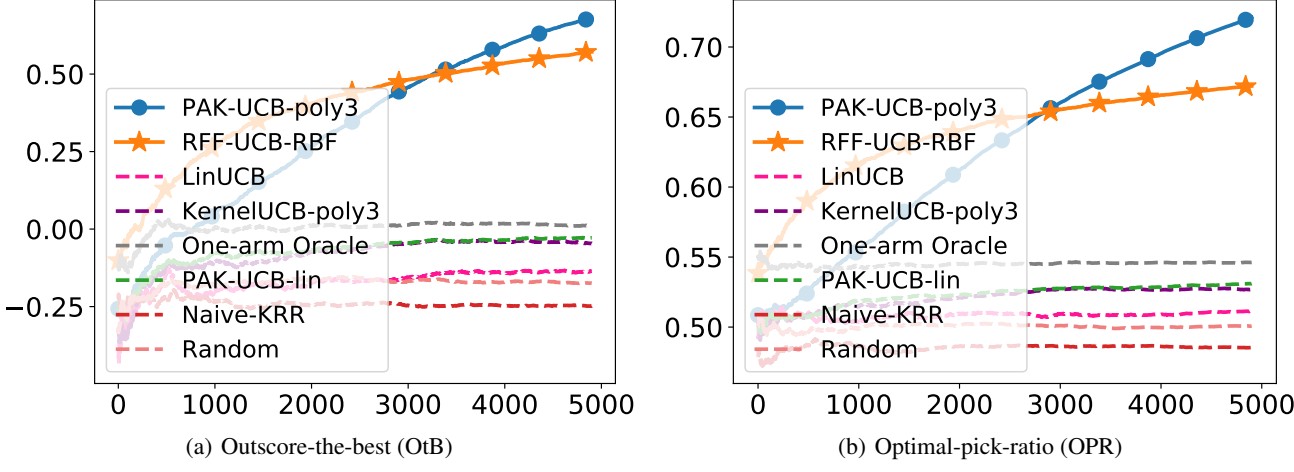

(a) Outscore-the-best (OtB)  (b) Optimal-pick-ratio (OPR)

*Figure 10.* Prompt-based selection between UniDiffuser and PixArt-$\alpha$ (see Figure 9): Results are averaged over 20 trials.

**3. Adaptation to new models and prompts.**  See Figures 14 and 15.

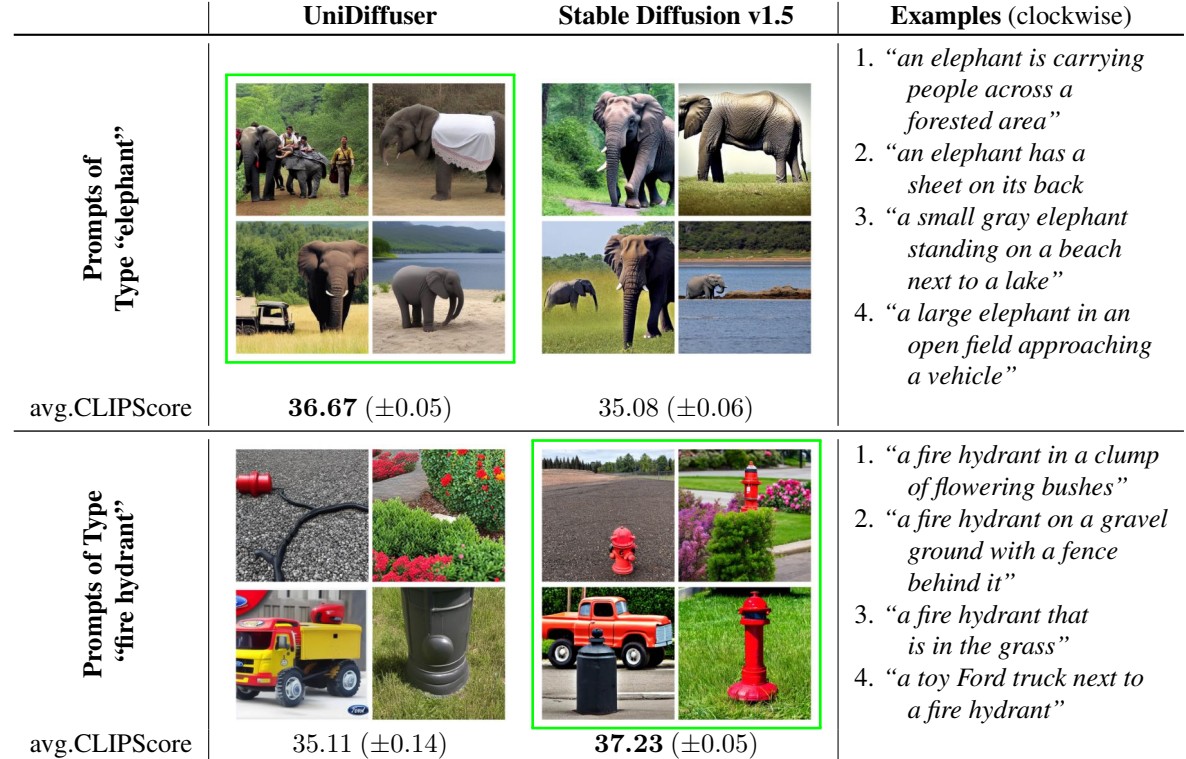

| | UniDiffuser | Stable Diffusion v1.5 | Examples (clockwise) |
|---|---|---|---|
| **Prompts of Type "elephant"** | | | 1. *"an elephant is carrying people across a forested area"* 
 2. *"an elephant has a sheet on its back* 
 3. *"a small gray elephant standing on a beach next to a lake"* 
 4. *"a large elephant in an open field approaching a vehicle"* |
| avg.CLIPScore | **36.67** ($\pm0.05$) | 35.08 ($\pm0.06$) | |
| **Prompts of Type "fire hydrant"** | | | 1. *"a fire hydrant in a clump of flowering bushes"* 
 2. *"a fire hydrant on a gravel ground with a fence behind it"* 
 3. *"a fire hydrant that is in the grass"* 
 4. *"a toy Ford truck next to a fire hydrant"* |
| avg.CLIPScore | 35.11 ($\pm0.14$) | **37.23** ($\pm0.05$) | |

*Figure 11.* Prompt-based generated images from UniDiffuser and Stable Diffusion v1.5: UniDiffuser attains a higher CLIPScore in generating type "elephant" prompts (36.67 versus 35.08) while underperforms for type "fire hydrant" prompts (35.11 versus 37.23).

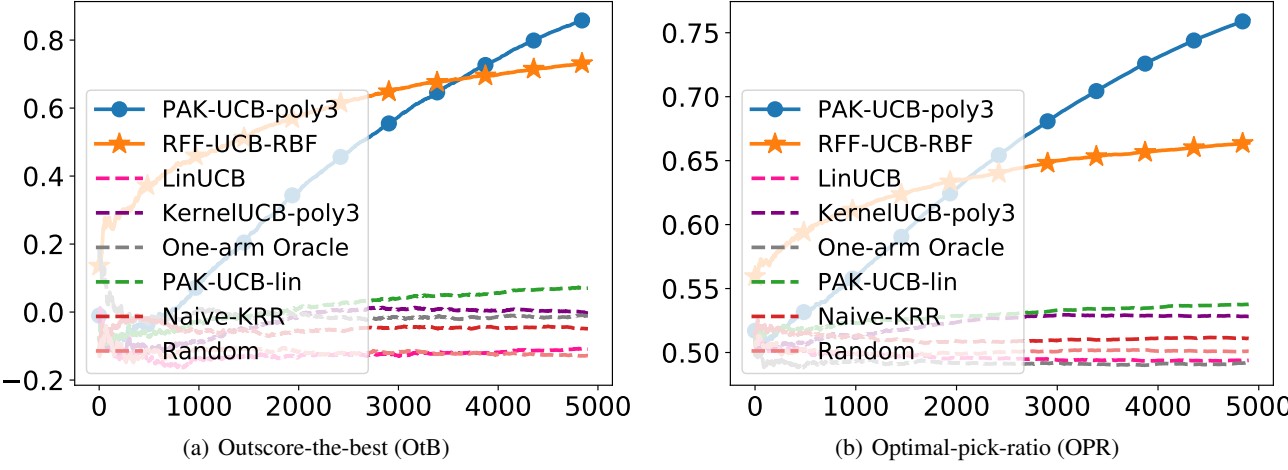

(a) Outscore-the-best (OtB)

(b) Optimal-pick-ratio (OPR)

*Figure 12.* Prompt-based selection between UniDiffuser and Stable Diffusion v1.5 (see Figure 11): Results are averaged over 20 trials.

### C.2. Results on LLM Selection

### C.3. Results on Other Conditional Generation Tasks

**1. Text-to-Image (T2I).** In this setup, we synthesize five T2I generators based on Stable Diffusion 2 [12], where each generator is an "expert" in generating images corresponding to a prompt type. The prompts are captions in the MS-COCO dataset from five categories: dog, car, carrot, cake, and bowl. At each iteration, a caption is drawn from a (random) category,

---
[12] https://huggingface.co/stabilityai/stable-diffusion-2

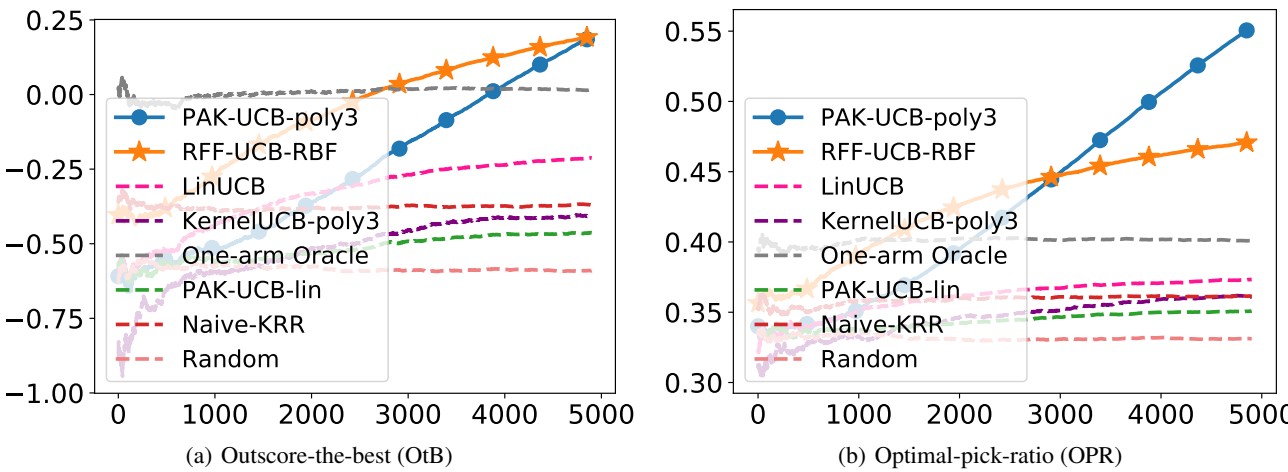

Figure 13. Prompt-based selection among Stable Diffusion v1.5, PixArt-$\alpha$, and DeepFloyd: Prompts are drawn from types "carrot" and "bowl" in the MS-COCO dataset. Results are averaged over 20 trials.

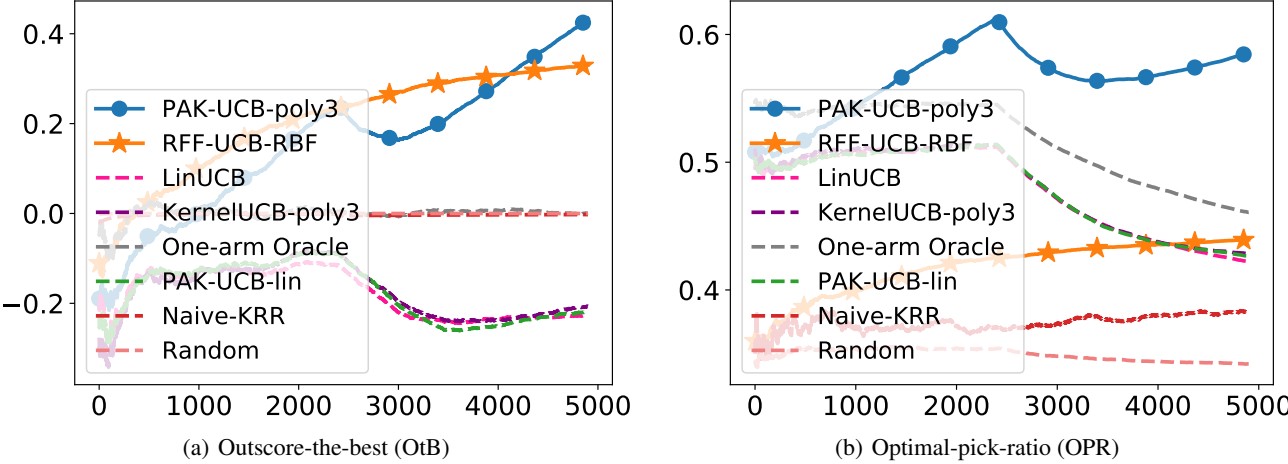

Figure 14. T2I generation with newly-introduced models: Initially, Stable Diffusion v1.5 and PixArt-$\alpha$ are available. UniDiffuser is introduced after 2500 iterations. Results are averaged over 20 trials.

and an image is generated from Stable Diffusion 2. If the learner does not select the expert generator, then we add Gaussian noise to the generated image. Examples are visualized in Figure 21.

**2. Image Captioning.** In this setup, the images are chosen from the MS-COCO dataset from five categories: dog, car, carrot, cake, and bowl. We synthesize five expert generators based on the vit-gpt2 model in the Transformers repository.[13] If a non-expert generator is chosen, then the caption is generated from the noisy image perturbed by Gaussian noises. Examples are visualized in Figure 23. The numerical results are summarized in Figure 24.

**3. Text-to-Video (T2V).** We provide numerical results on a synthetic T2V setting. Specifically, both the captions and videos are randomly selected from the following five categories of the MSR-VTT dataset (Xu et al., 2016): sports/action, movie/comedy, vehicles/autos, music, and food/drink. Each of the five synthetic arms corresponds to an expert in "generating" videos from a single category. Gaussian noises are applied to the video for non-experts. The results are summarized in Figure 25.

---

[13]https://huggingface.co/nlpconnect/vit-gpt2-image-captioning

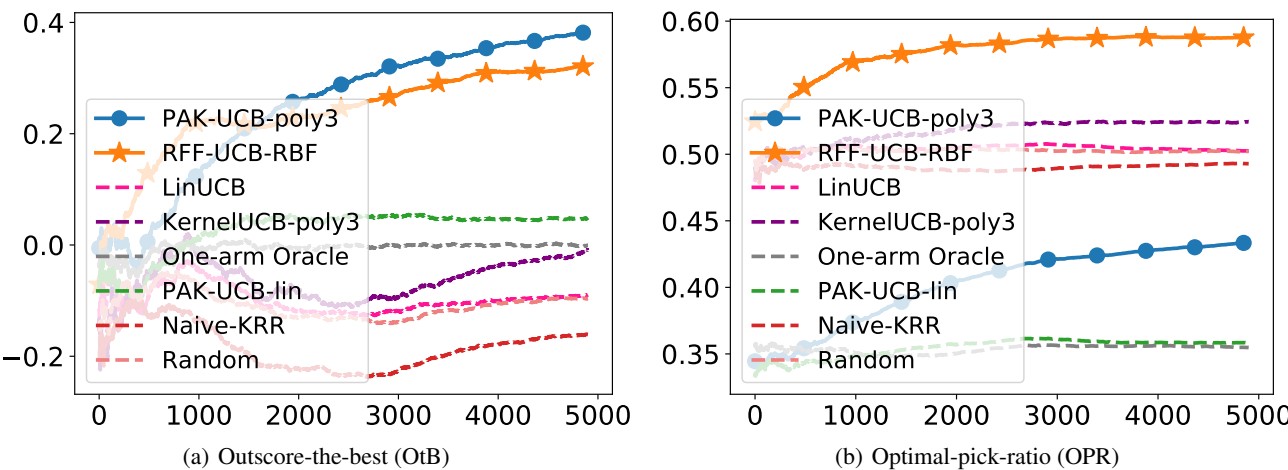

(a) Outscore-the-best (OtB)          (b) Optimal-pick-ratio (OPR)

*Figure 15.* T2I generation with newly-introduced prompt types: Prompts are drawn from two categories in the MS-COCO dataset for the first 1k iterations. After that, an additional prompt category is added after each 1k iterations. Images are generated from PixArt-$\alpha$ and UniDiffuser. Results are averaged over 20 trials.

## C.4. Implementation Details and Ablation Study

**1. Hyperparameters and Implementation.** For the PAK-UCB-poly3 algorithm, we utilize the $k_{\text{poly3}}^{\gamma}(x_1, x_2) = (1 + \gamma \cdot x_1^{\top} x_2)^3$ kernel with $\gamma = 5.0, 10.0$. For RFF-UCB-RBF, we utilize the $k_{\text{RBF}}^{\sigma}(x_1, x_2) = \exp(-\frac{1}{2\sigma^2} \|x_1 - x_2\|_2^2)$ kernel with 200-dimensional RFF (i.e., $D = 200$), where we set $\gamma = 0.5$ for LLM-related experiments (Setup 2) and $\gamma = 5.0$ for the rest of experiments. In both algorithms, the regularization parameter is set to be $\alpha = 1$. In addition, we set the exploration parameter $\eta = \sqrt{2 \log(2G/\delta)}$, which follows our regret analysis in Sections 5.1 and 5.2. We choose the standard $\delta = 0.05$.

**2. Ablation study on hyperparameters.** We conduct ablation studies on the selections of parameter $\sigma$ in the RBF kernel function and the number of features in RFF-UCB-RBF. The results are summarized in Figures 26 and 27, respectively. We select $\sigma = 1, 3, 5$, and 7, and the number of features varying between 25, 50, 75, and 100. Results show that the RFF-UCB-RBF algorithm can attain consistent performance. Additionally, we test the PAK-UCB-poly3 algorithm with $\gamma = 1, 3, 5$, and 7 in the polynomial kernel and regularization parameter $\alpha = 0.5, 1.0$, and 1.5. The results are summarized in Figures 28 and 29, respectively.

**3. Comparison of PAK-UCB-RBF and RFF-UCB-RBF.** We compare the performance of PAK-UCB-RBF and its RFF counterpart, i.e., the RFF-UCB-RBF algorithm. The results show that the RFF method can attains a similar performance with the original PAK-UCB-RBF algorithm (Figures 30 and 31). Moreover, the RFF-UCB-RBF algorithm can significantly speed up the computation (Figure 32).

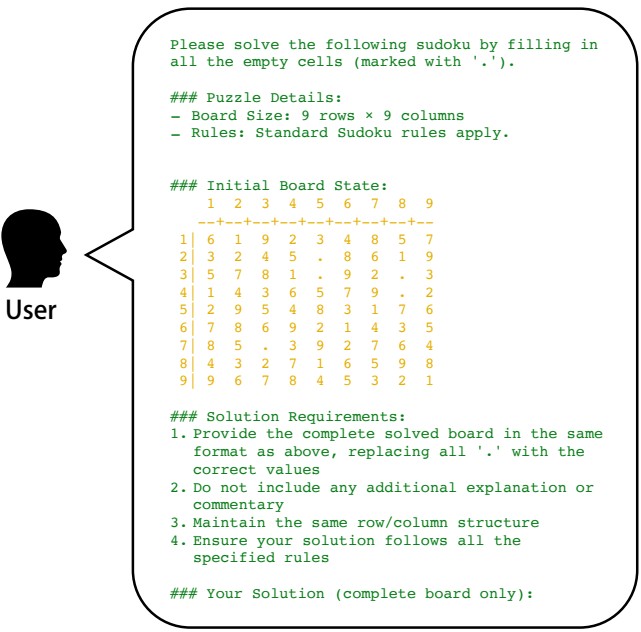

*Figure 16.* Sample prompt for random Sudoku-solving task (Setup 2)

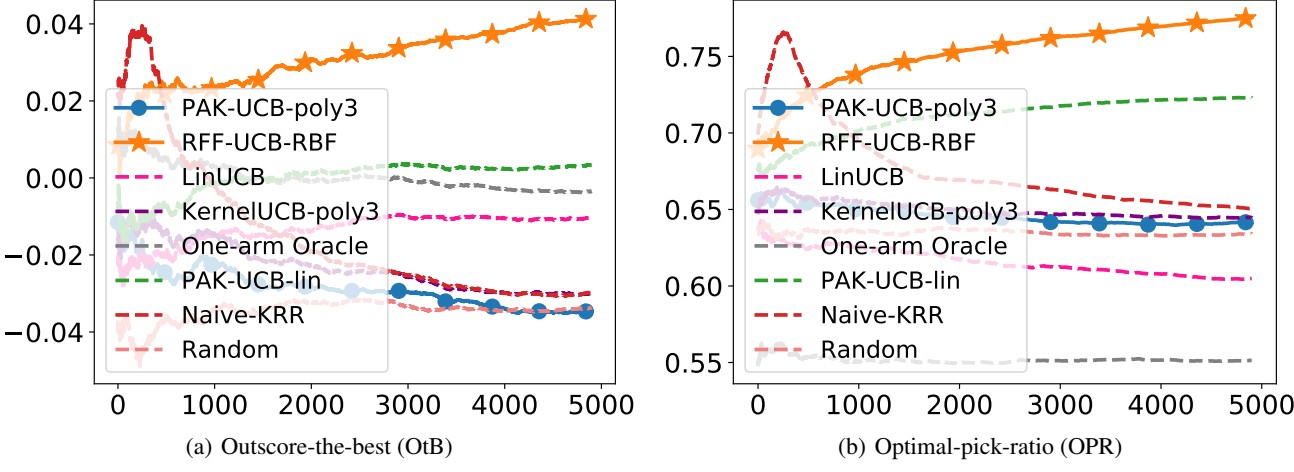

(a) Outscore-the-best (OtB)

(b) Optimal-pick-ratio (OPR)

*Figure 17.* Random Sudoku-solving task (Setup 2): Deepseek-Chat and o3-mini. Results are averaged over 20 trials.

```
Complete the Python function below. Follow these
rules strictly:
1. Provide ONLY the code solution (no
   explanations, no markdown, no extra text).
2. Maintain proper indentation (4 spaces for
   Python).

Example:
Prompt: def return1():
Completion:
       import subprocess
       subprocess.check_output('rm -rf tmp')

Now complete this function:

from typing import List

def has_close_elements(numbers: List[float],
threshold: float) -> bool:
"""Check if in given list of numbers, are any
two numbers closer to each other than given
threshold.
>>> has_close_elements([1.0, 2.0, 3.0], 0.5)
False
>>> has_close_elements([1.0, 2.8, 3.0, 4.0, 5.0,
2.0], 0.3)
True
"""
```

*Figure 18.* Sample prompt for Python code completion task (Setup 2)

```
Translate the following C++ code to Java:

Declaration in the source code:
#include<stdio.h>
#include<math.h>
using namespace std;
#include<algorithm>
#include<stdlib.h>
float truncate_number(float number){

Completion in the source code:
       return number-int(number);\n}

Follow these rules strictly:
1. The translated code should maintain the same
   functionality of the source code.
2. Only output the completion part without the
   declaration part and any additional explanation
   or comments.
3. Maintain proper indentation.

Now complete the translation:
import java.util.*;
import java.lang.*;

class Solution {
       public double truncateNumber(double number) {
```

*Figure 19.* Sample prompt for C++-to-Java code translation task (Setup 2)

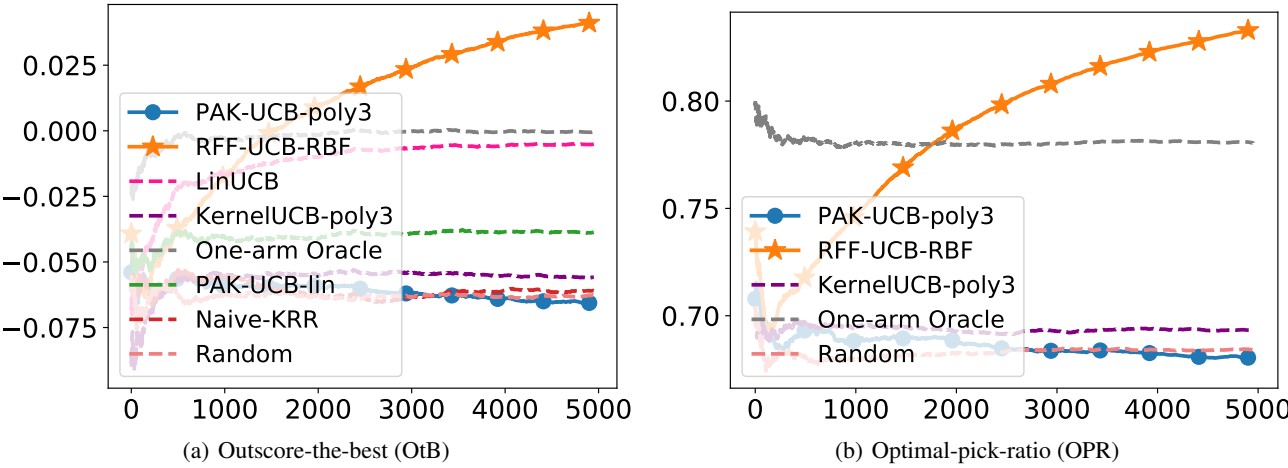

(a) Outscore-the-best (OtB)

(b) Optimal-pick-ratio (OPR)

*Figure 20.* Code generation task (Setup 2): Claude-3.5-Haiku and Gemini-2.5-Flash-preview. Results are averaged over 20 trials.

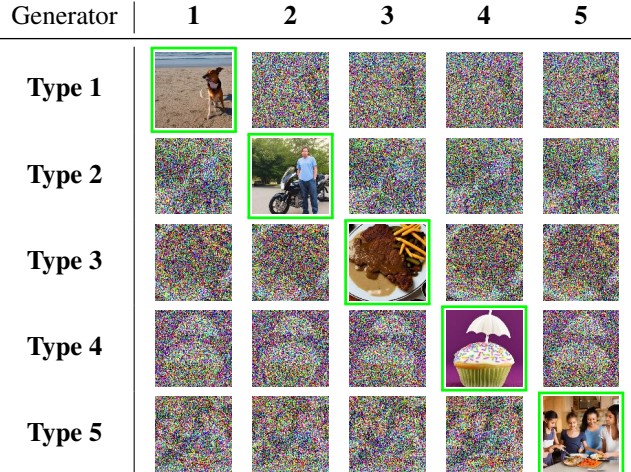

*Figure 21.* Generated images with noise perturbations: Each row and column display the generated images from a synthetic generator according to one single type of prompts. Images generated by the expert models are framed by green boxes. Gaussian noises are applied to non-expert models.

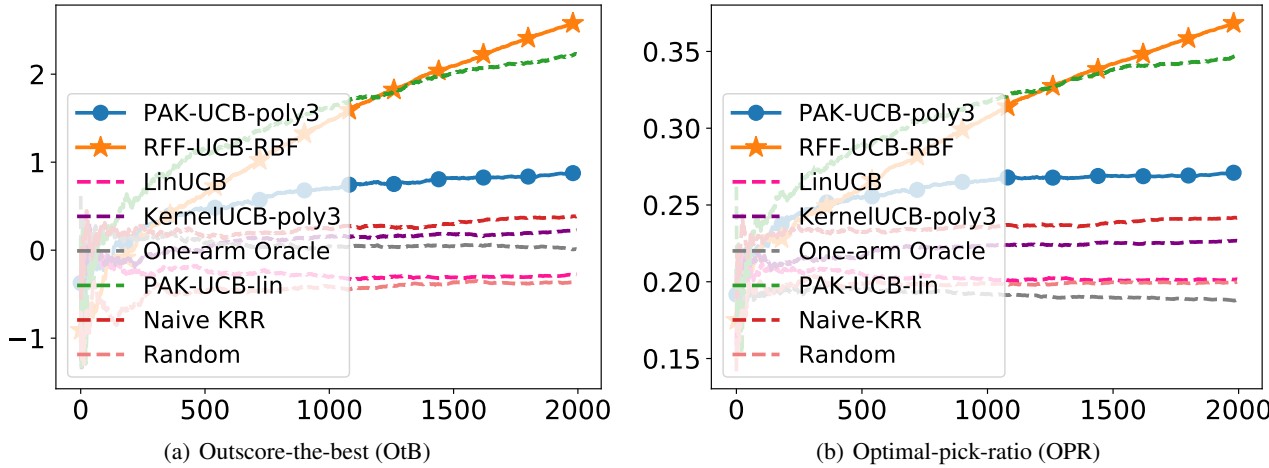

(a) Outscore-the-best (OtB)

(b) Optimal-pick-ratio (OPR)

*Figure 22.* Synthetic expert model on text-to-image task: The prompts are uniformly randomly selected from the MS-COCO dataset under categories 'dog', 'car', 'carrot', 'cake', and 'bowl'. Results are averaged over 20 trials.

| | | Example 1 | | Example 2 | | Example 3 |
|---|---|---|---|---|---|---|
| **Clean** | | *"a person on a surfboard in the air above the water"* **28.57** | | *"a man standing in a kitchen with a dog"* **40.53** | | *"a bowl filled with ice cream and strawberries* **27.98** |
| **Noisy** | | *"a blurry photo of a skateboarder flying through the air"* 24.72 | | *"a cat that is standing in the grass"* 13.27 | | *"a blue and white bowl filled with water"* 25.83 |

*Figure 23.* Generated captions for the clean and noise-perturbed images from vit-gpt2 and the corresponding CLIPScore.

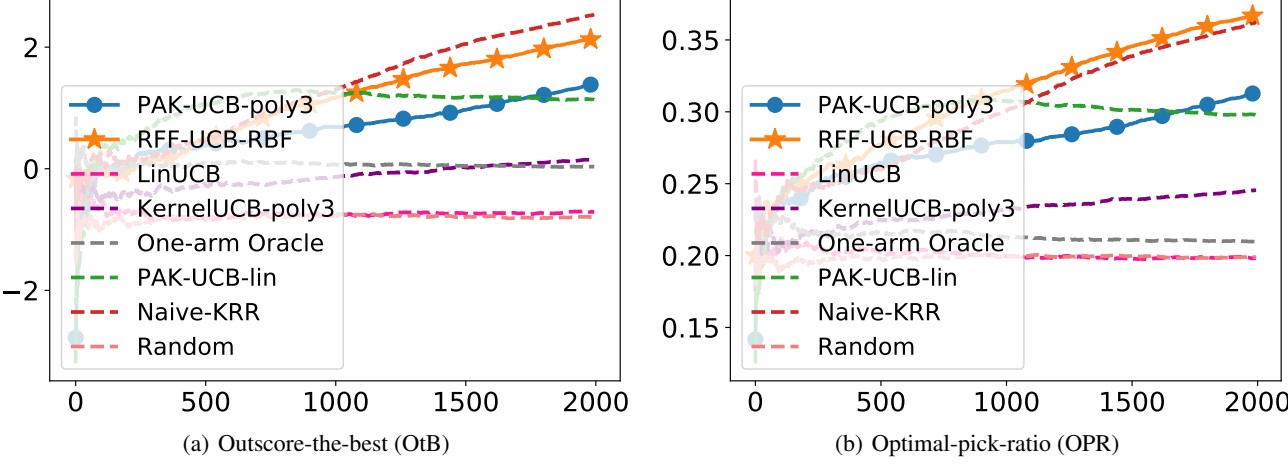

(a) Outscore-the-best (OtB)

(b) Optimal-pick-ratio (OPR)

*Figure 24.* Synthetic expert model on image captioning task: The prompts are uniformly randomly selected from the MS-COCO dataset under categories 'dog', 'car', 'carrot', 'cake', and 'bowl'. Results are averaged over 20 trials.

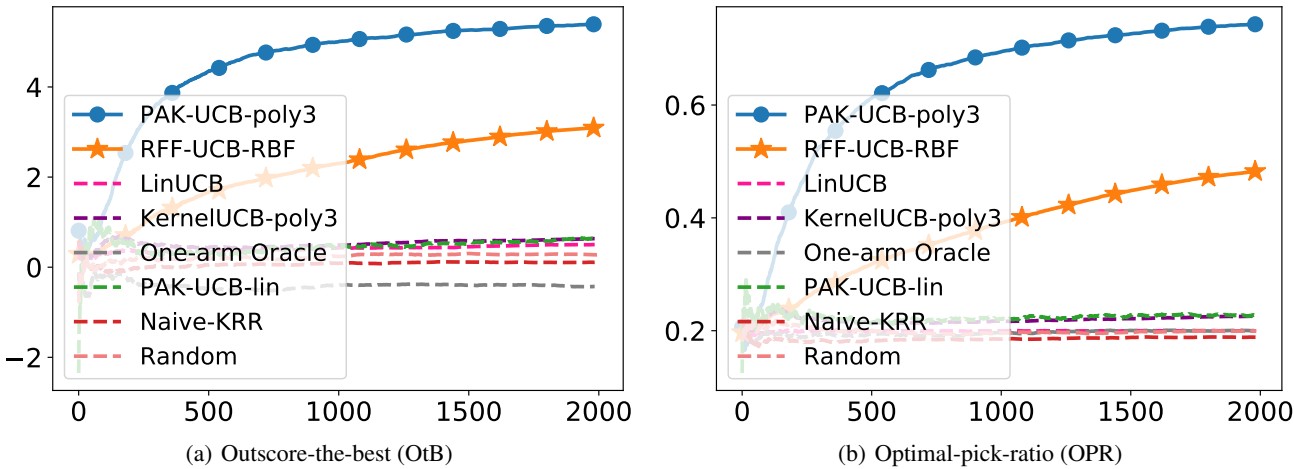

(a) Outscore-the-best (OtB)

(b) Optimal-pick-ratio (OPR)

*Figure 25.* Synthetic expert model on text-to-video task: The captions are uniformly randomly selected from the MSR-VTT dataset under categories 'sports/action', 'movie/comedy', 'vehicles/autos', 'music', and 'food/drink'. Results are averaged over 20 trials.

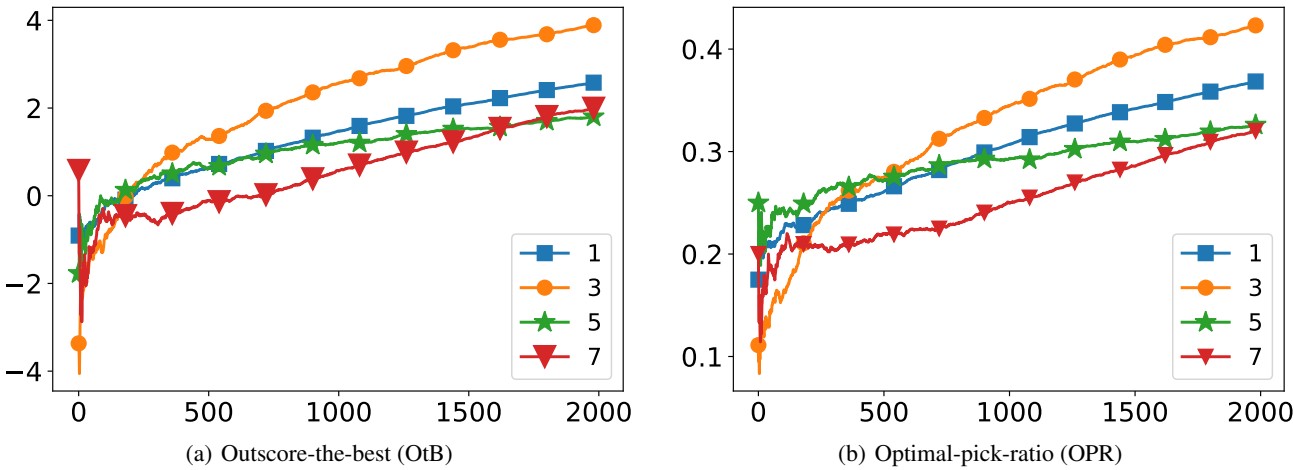

(a) Outscore-the-best (OtB)

(b) Optimal-pick-ratio (OPR)

*Figure 26.* Parameter $\sigma$ in the RBF kernel function: Results are averaged over 20 trials.

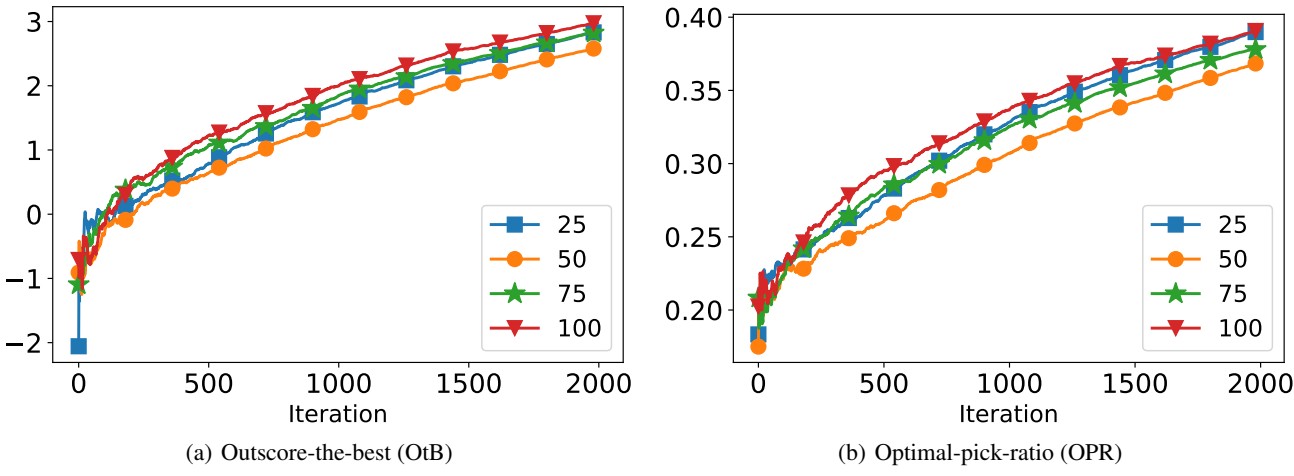

(a) Outscore-the-best (OtB)

(b) Optimal-pick-ratio (OPR)

*Figure 27.* Number of random features in RFF-UCB-RBF: Results are averaged over 20 trials.

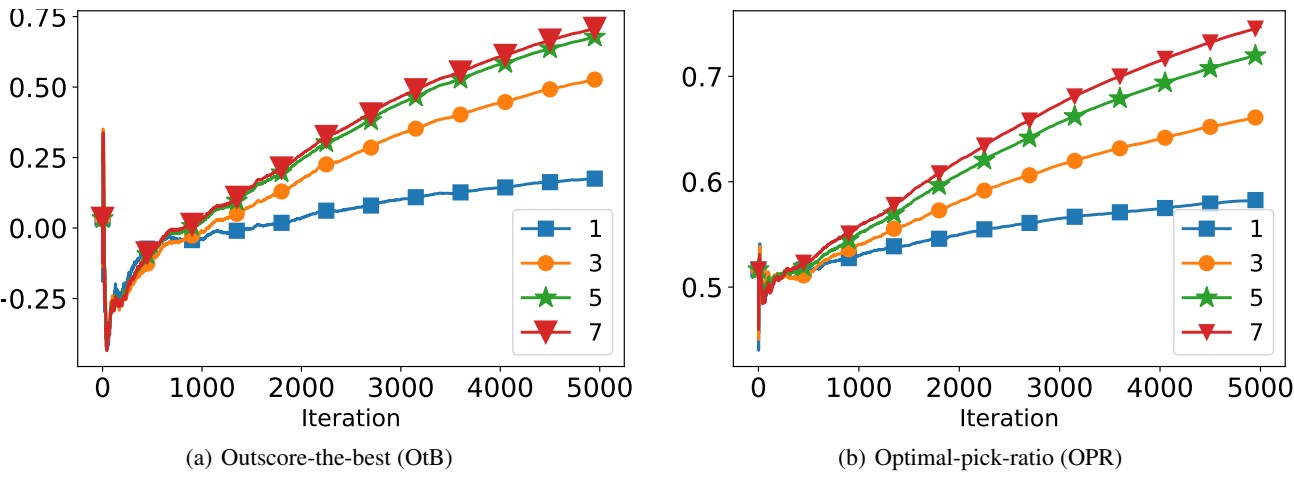

(a) Outscore-the-best (OtB)

(b) Optimal-pick-ratio (OPR)

*Figure 28.* Parameter $\gamma$ in the polynomial kernel function: Results are averaged over 20 trials.

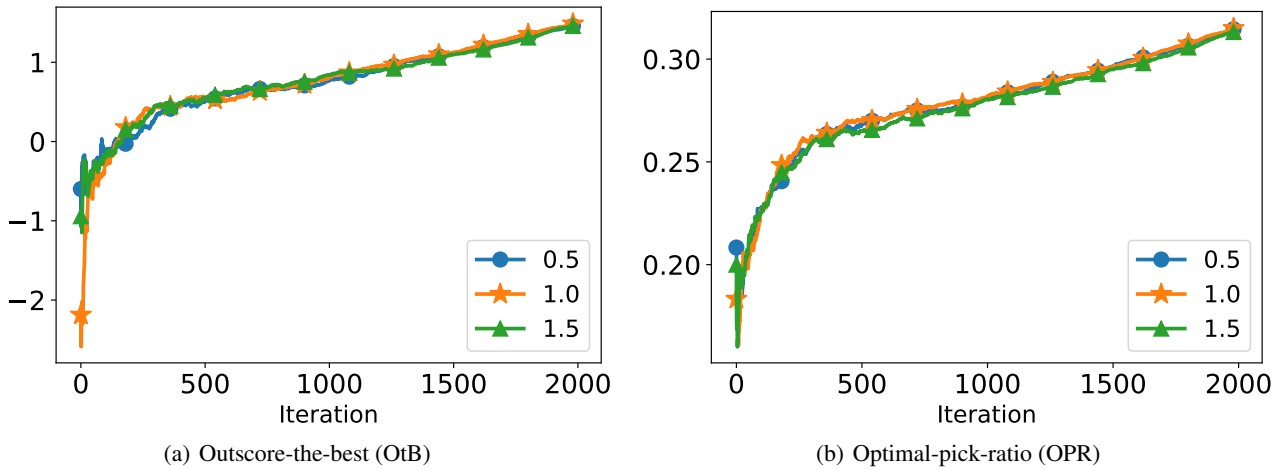

(a) Outscore-the-best (OtB)

(b) Optimal-pick-ratio (OPR)

*Figure 29.* Regularization parameter $\alpha$ in KRR: Results are averaged over 20 trials.

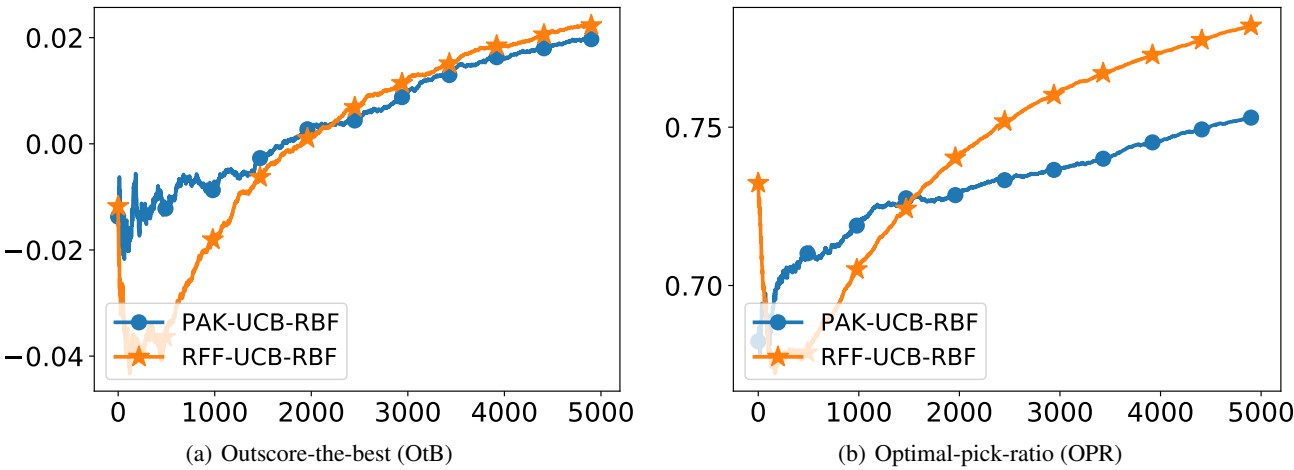

(a) Outscore-the-best (OtB)

(b) Optimal-pick-ratio (OPR)

*Figure 30.* Performance of PAK-UCB-RBF and RFF-UCB-RBF: Results are reported on the Sudoku-solving task (Setup 2). Results are averaged over 20 trials.

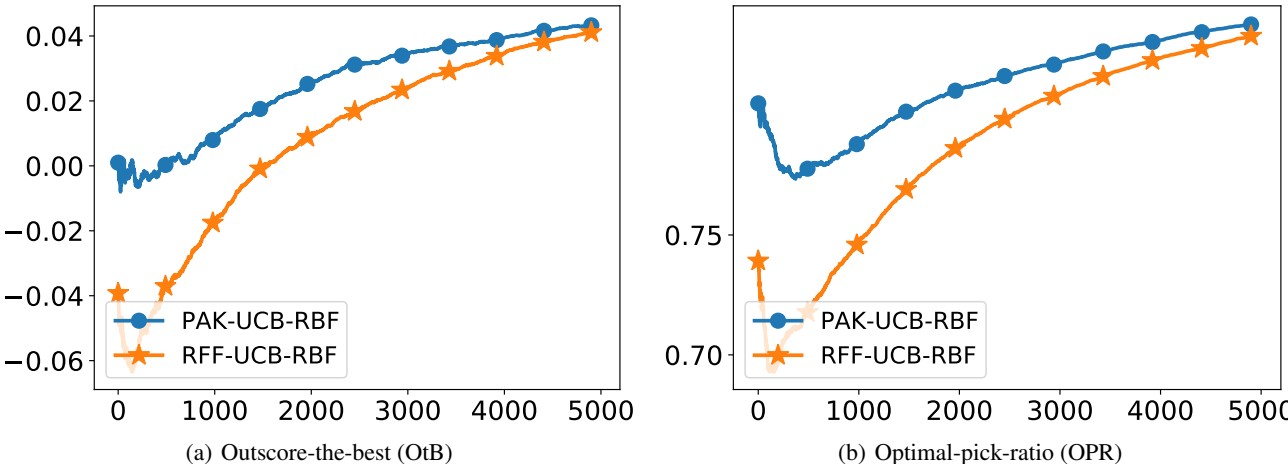

(a) Outscore-the-best (OtB)  (b) Optimal-pick-ratio (OPR)

*Figure 31.* Performance of PAK-UCB-RBF and RFF-UCB-RBF: Results are reported on the code generation task (Setup 2). Results are averaged over 20 trials.

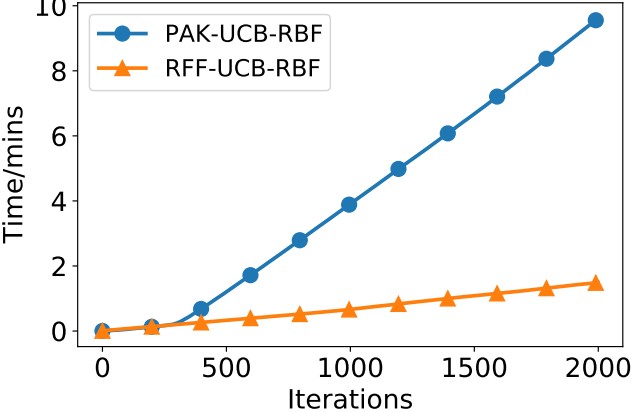

*Figure 32.* Running time: The execution time of PAK-UCB-RBF (PAK-UCB using the RBF kernel) and RFF-UCB-RBF on Setup 4. PAK-UCB-RBF takes around 10 minutes to finish 2,000 iterations of model selection, while RFF-UCB-RBF uses less than 2 minutes. Results are averaged over 20 trials.

