# OpenReview forum: "PAK-UCB Contextual Bandit: An Online Learning Approach to Prompt-Aware Selection of Generative Models and LLMs"
_ICML.cc/2025/Conference — ICML 2025 poster_

### Official Review · Reviewer_5QGu · 2025-03-09

**Overall Recommendation:** 3

**Summary:**

- This paper frames the task of optimally routing prompt to a data generation model as a contextual bandit problem
-  In doing so, the authors design a contextual bandit algorithm called PAK-UCB and prove uapper bounds on its expected regret
-  The overcome the computational overhead of PAK-UCB, the authors propose a variant, called RFF-UCB, based on the random Fourier Features framework that approximates PAK-UCB. They prove that RFF-UCB is efficient and obtains expected regret that is not too much larger than that of PAK-UCB.
- Finally, the authors present experimental results showcasing the performance of PAK-UCB and RFF-UCB for text-to-image and image-captioning tasks.

## Update after rebuttal
I thank the authors for their response. As they have satisfactorily addressed my questions and concerns, I will maintain my positive score for this paper.

**Claims And Evidence:**

Yes, the claims made in the submissions are supported by clear and convincing evidence.

**Essential References Not Discussed:**

From what I can tell, the authors adaquetly cite existing work and make sure to distinguish their settings with existing work.

**Experimental Designs Or Analyses:**

Yes, I checked the experimental results in the main text and the Appendix.

**Methods And Evaluation Criteria:**

Yes, the proposed methods and/or evaluation criteria make sense for the problem at hand.

**Other Comments Or Suggestions:**

- I think the authors should move Remark 1 into the Contextual bandits section of the Related works to futher drive home the differences between their setup and the standard CB setup
- In the Equation in Lines 212-213, $\tilde{\Phi}$  and $\tilde{\Phi}^*$ are not defined any where.
- The authors provide theoretical guarantees on regret, but the experiments are evaluted with respect to O2B and OPR. I would be interested in seeing plots on regret as well.

**Other Strengths And Weaknesses:**

**Strengths**:
- This paper is well written and easy to follow
- The problem studied is well-motivated and the novel variant of the CB problem is interesting to me
- The experimental results for PAK-UCB seem fairly strong to me

**Weaknesses**:
- Clarity.  The regret guarantees of PAK-UCB are with respect to a variant of the PAK-UCB algorithm presented by the authors in the main text, with no discussion on what this variant is and why it is needed. Moreover, it now becomes unclear whether the xperimental results for PAk-UCB in Section 6 are with respect to Algorithm 2 or its variant in Appendix A.1. The same can be said about RFF-UCB. In fact, its unclear whether Lemma 2 still holds the variant of RFF-UCB that satisfies Theorem 2.
- Mismatch between theory and practice. According to Theorem 1 and Theorem 2, the reader gets the sense that PAK-UCB and RFF-UCB have the same regret bound and thus should perform empirically similary. In fact, the authors state "It can be shown that the implementation of PAK-UCB with RFF attains the exact same regret guarantees for adaptively selected feature sizes." However, the experiments tell a different story, with PAK-UCB consistently and significantly outperforming RFF-UCB across all experimental setups. Moreover, in many of these experiments, RFF-UCB doesn't seem to significantly outperform the baselines. It would be nice if the authors can provide some reasonable justification about why RFF-UCB doesn't do as well as PAK-UCB despite having similar regret guarantees.

**Questions For Authors:**

(1) What are lower bounds on expected regret in your variant of the CB problem? Are your algorithms optimal with respect to the relevant problem parameters?

**Relation To Broader Scientific Literature:**

This paper empirically finds that different models perform better on different prompts. This motivates the design of routing mechanisms which can sequentially learn to route prompts to model so that prompts get answered by their "optimal" model. The authors frame this problem through as a contextual bandit problem, where now the context is the prompt, and the arms are the different models that the prompt can be routed to. Unlike the standard contextual bandit setting where a different context vector is variable is observed for each arm, but a single weight vector is applied to all arms, in the authors setting, the flip is true: there is a single context that is shared across all arms, but now each arm has a fixed weight vector.  As far as I can tell (I am not an expert in this area), both the framing of prompt-based selection of generative models as a variant of the CB problem and the particular variant of the CB problem are novel, and worth futher investigations.  As such, a key contribution of this paper is the introduction of a new variant of the CB problem and its applications to modern-day generative machine learning.

**Theoretical Claims:**

As no proofs were provided in the main text, I did not verify the correctness of the theoretical claims.

---

> ### Author Rebuttal · Authors · 2025-03-31
>
> We thank Reviewer 5QGu for the thoughtful feedback on our work. Below is our answer to the reviewer's comments and questions.
>
>
> **1. Regret and complexity of PAK-UCB and RFF-UCB**
>
> First, we would like to clarify that the numerical results in Section 6 are reported for PAK-UCB (Alg.2) and RFF-UCB (Alg.3) in the main text.
>
> Also, as we stated in Theorem 1, the regret bound is shown for the variant of PAK-UCB in the Appendix, which we titled Sup-PAK-UCB (Alg.4). We note that our analysis of PAK-UCB and Sup-PAK-UCB parallels the analysis of LinUCB [1] and KernelUCB [2] references. Specifically, [1] introduces two versions of the LinUCB algorithm: "LinUCB" (Alg.1 on p.g. 3 in [1]) is recommended for usage in practice, whereas [1]'s' regret analysis is performed for "SupLinUCB" (Alg.3 on p.g. 4). Similarly, Reference [2] also introduces two versions of KernelUCB, "KernelUCB" (Alg.1 on p.g. 5) and "SupKernelUCB" (Alg.2 on p.g. 5), where the numerical application and theoretical analysis are aimed for the algorithms. Our analysis also follows a similar approach. We will include this clarification in the revised paper.
>
> **2. Regarding Lemma 2**
>
> We would like to clarify that Lemma 2 still holds for the variant of RFF-UCB analyzed in Theorem 2 (described in Appendix B.2). Note that Sup-PAK-UCB (Alg.4) computes the UCB values on the mutually exclusive subsets $\\{\Psi\_g^m\\}\_{m \in [M]}$, which satisfies $\sum_{m,g}|\Psi_g^m| \le t$ at the $(t+1)$-th iteration. Therefore, Sup-PAK-UCB using Compute_UCB_RFF (Alg.3) requires time at most $\Theta(\sum_{m,g}|\Psi_g^m| s^2) = O(ts^2)$ and space $\Theta(\sum_{m,g} |\Psi_g^m| s) = O(ts)$. We will add this clarification to the revised paper.
>
> **3. Performance of RFF-UCB**
>
> As pointed out by the reviewer, there could be a performance gap between PAK-UCB-poly3 and RFF-UCB (e.g. in Figure 2). Note that this is not in contradiction with the regret bound of Theorem 2, because PAK-UCB-poly3 and RFF-UCB use *different kernel functions* to predict the scores: RFF-UCB uses the Gaussian kernel, whereas PAK-UCB-poly3 uses a polynomial kernel with degree 3. Therefore, RFF-UCB is *not* supposed to match the result with PAK-UCB-poly3.
>
>
> **3. Discussing Remark 1 in the Related work**
>
> Thank you for the suggestion. We will update the related work with a summary of Remark 1.
>
>
> **4. Typo in lines 212-213**
>
> We thank the reviewer for pointing this out. We would like to clarify that the tilde in the notation $\widetilde{\Phi}$ is a typo, which we will correct in the revision. In the equation, $\widetilde{\Phi}$ should be replaced with $\Phi$.
>
>
> **5. Results on regret**
>
> We appreciate the reviewer's question about the numerical performance in terms of regret values. We note that the evaluations based on O2B and the (average) regret are equivalent. This is because the *average regret* is computed as $\text{Avg.Regret}(T) := \frac{1}{T} \sum_{t=1}^T (s_\star (y_t) - s_{g_t}(y_t))$, while *O2B* is computed as $$\text{O2B}(T) := \frac{1}{T}\sum_{t=1}^T (s_{g_t}(y_t) - s_{g^\star}(y_t)) = -\text{Avg.Regret}(T) + C,$$ where $g^\star$ is the best single model with the highest expected score. Therefore, the O2B scores of different policies are all shifted with the same constant $C := \frac{1}{T}\sum_{t=1}^T (s_\star (y_t) - s_{g^\star}(y_t))$ compared to their average regret. As a result, the O2B rankings of the policies in the plots are *identical* to the regret-based rankings.
>
>
> **6. Regret lower bound**
>
> First, we note that the regret (upper) bound derived in Theorem 1 matches the regret of the LinUCB [1] and KernelUCB [2] algorithms up to a factor of $\sqrt{G}$, where $G$ is the number of models. On the other hand, we anticipate that the regret lower bound could scale with $\Omega(\sqrt{dGT})$ for a kernel function with a finite dimension $d$ (e.g., by slight modification of [Theorem 2, 1] for linear bandits without arm-specification). Formally proving a regret lower bound for the arm-specific setting is an interesting future direction for our work. We will discuss this in the revised conclusion.
>
>
> [1] Chu, et al. "Contextual bandits with linear payoff functions." AISTATS 2011.
>
> [2] Valko, et al. "Finite-time analysis of kernelised contextual bandits", UAI 2013.

---

> > ### Comment · Reviewer_5QGu · 2025-04-04
> >
> > I thank the authors for their response and for addressing my questions and concerns. I will maintain my positive score.

---

> > > ### Author Response · Authors · 2025-04-05
> > >
> > > We would like to thank Reviewer 5QGu for the constructive review and feedback on our response. We are pleased that our response helped address the reviewer’s concerns. As noted, we will revise the paper accordingly to incorporate the discussed improvements.

---

### Official Review · Reviewer_3Dpj · 2025-03-17

**Overall Recommendation:** 3

**Summary:**

Generative models are increasingly being used in numerous applications. Evaluation scores are typically used when selecting a sample generation from multiple models. The drawback of evaluation scores is that different models perform better under different text prompts. The paper proposes a method to address this issue by learning the ranking of generative models for a given prompt.

The proposed method goes beyond standard LinUCB and KernelUCB. Specifically, it introduces PAK-UCB, which learns an arm-specific function to predict the score of each model. Furthermore, the paper seeks to reduce expensive computation and memory overhead by incorporating Random Fourier Features into PAK-UCB. The proposed algorithm is evaluated against several baselines for prompt-based selection of text-to-image models and wa shown to outperform all of them.

**Claims And Evidence:**

The paper claims that different models perform differently depending on the text prompts provided, and evaluation scores do not capture this inherent limitation. The authors present evidence of this through their experiments, which is well-known in the literature. They propose to address this issue by converting the problem of finding which model responds best to which prompt—maximizing evaluation scores—into a contextual bandit problem.

**Essential References Not Discussed:**

None

**Experimental Designs Or Analyses:**

The paper conducts experiments using state-of-the-art models: UniDiffuser, Stable Diffusion, PixelArt, and DeepFloyd. Specifically, it evaluates the performance of these models using two metrics: (i) outscore-the-best (02B) and (ii) optimal-pick-ratio (OPR). The main results of the paper are sound, and the experimental setup is well-defined. However, one issue is that while the models used were provided, the paper does not mention any of the datasets that were used in the experiments.

In addition to the main results, the paper includes two additional ablation studies—one for adaptation to new prompts and the other for synthetic experiments. Similar to the main results, the authors do not clarify how the new prompts were selected or what the original datasets were that the models were trained on. Furthermore, the paper does not explain the overlap between the new prompts and the training prompts. The synthetic experiments also lack detail, making it unclear how these experiments were conducted.

**Methods And Evaluation Criteria:**

Yes, the proposed methods and evaluation make sense for the problem and application.

**Other Comments Or Suggestions:**

None

**Other Strengths And Weaknesses:**

None

**Questions For Authors:**

1) What dataset did you perform your experiments on?

2) What evaluation score (e.g., ClipScore) did you use in your main experiments?

3) What features are you using for your kernel methods?

4) How does randomly selecting a model per prompt perform?

5) Are non-polynomial degree 3 algorithms suffering from the model not being expressive enough?

6) Could you elaborate on the difference between PAK-UCB and KernelUCB?

**Relation To Broader Scientific Literature:**

The authors are attempting to address an ongoing and challenging problem: models are sensitive to prompts, and small changes in the prompts can have significant effects on the model's decision-making. In particular, the authors are trying to improve the selection criteria for determining which model should be used for a given prompt in order to maximize performance.

**Theoretical Claims:**

The paper claims that the proposed algorithm achieves a \bigO\sqrt{GT} regret bound. I briefly reviewed the proof for correctness.

---

> ### Author Rebuttal · Authors · 2025-03-31
>
> We thank Reviewer 3Dpj for the thoughtful feedback on our work. Below is our answer to the reviewer's comments and questions.
>
>
> **1. Details of the datasets**
>
> We note that the details of experiment settings are discussed in Appendix D. The following is a summary of the details asked by Reviewer 3Dpj, which we will include in the revised main text to improve the clarity:
>
> * **Setup 1:** Prompts are uniformly randomly selected from the MS-COCO dataset under two categories: 'dog'/'car' (Fig.2), 'train'/'baseball-bat' (Fig.3a),  'elephant'/'fire-hydrant' (Fig.3b), and 'carrot'/'bowl' (Fig.3c).
> * **Adaptation to new models (Fig.4a):** Prompts are uniformly randomly selected from the MS-COCO dataset under categories 'train' and 'baseball-bat',
> * **Adaptation to new prompt types (Fig.4b):** In the first 1k iterations, the prompts are uniformly randomly selected from a pool that initially includes categories 'person' and 'bicycle' in the MS-COCO dataset. Then, categories 'airplane', 'bus', 'train', and 'truck' are added to the pool after each 1k iterations,
> * **Synthetic T2I and image-captioning experiments (Fig.19 and Fig.21):** The prompts are uniformly randomly selected from the MS-COCO dataset under categories 'dog', 'car', 'carrot', 'cake', and 'bowl',
> * **Synthetic T2V task (Fig.22):** The captions are uniformly randomly selected from the MSR-VTT dataset under categories 'sports/action', 'movie/comedy', 'vehicles/autos', 'music', and 'food/drink'.
>
>
> **2. Evaluation scores**
>
> In the numerical experiments of this paper, we primarily focus on text-to-image generation tasks and use CLIPScore as the evaluation score. We note that the online selection framework can be applied to other prompt-guided generation tasks as long as we know the score values assigned to the generated samples.
>
>
> **3. Features for kernel methods**
>
> The input of the PAK-UCB method and other baselines is the embedded prompt that is output by the pretrained CLIP-ViT-B-32-laion2B-e16 model from the open_clip repository (https://github.com/mlfoundations/open_clip/tree/main). Only for LinUCB and KernelizedUCB baselines, we also concatenate the one-hot encoded vector of the model index to the CLIP-embedded prompt.
>
> **4. Performance of the random selection strategy**
>
> We thank the reviewer for the suggestion of including the random selection strategy as a baseline. The random selection strategy would be expected to underperform for prompt-based model selection. Below, we provide the results in Setup 1 (Figure 3b) where we use PAK-UCB-poly3 as the competing strategy.
>
> | Metric (After 5k iterations) | Random | PAK-UCB-poly3 |
> | - | - | - |
> | **O2B** | -0.13 | **0.86** |
> | **OPR** |  0.50 | **0.77** |
>
> **5. Expressivity of score estimation functions**
>
> The use of a polynomial kernel with degree 3 is inspired by the kernel Inception distance (KID) in the literature of generative models [1, 2]. Please note that a higher degree can lead to higher expressivity while increasing the risk of overfitting the data. Below, we conduct an ablation study to test the effect of degree on the performance of the PAK-UCB algorithm using a polynomial. We observe that a degree of 3 can achieve a better tradeoff between expressivity and generalization.
>
> |Metric (After 5k iterations)|poly1|poly2|poly3|poly 4|
> |-|-|-|-|-|
> | **O2B** | 0.13 | 0.30 | **0.70** | 0.39 |
> | **OPR** | 0.54 | 0.58 | **0.71** | 0.61 |
>
> [1] Stein et al. "Exposing flaws of generative model evaluation metrics and their unfair treatment of diffusion models." NuerIPS 2023.
>
> [2] Bińkowski et al. "Demystifying mmd gans." ICLR 2018.
>
> **6. Comparison to KernelUCB**
>
> In the following, we will compare the process in PAK-UCB and KernelUCB side-by-side to highlight their differences:
>
> - **Problem Setting of PAK-UCB (ours):** We have $G$ arms where each arm represents a *fixed generative model* that remains unchanged across rounds: for example, Arm 1 represents the Stable Diffusion model in all rounds. At each round, the arms observe *one shared* context variable (i.e., the text prompt). We learn $N$ *separate kernel-based models* with *different weights* to predict the CLIPScore of a shared incoming prompt (context) for the $G$ fixed generative models.
> - **Problem Setting of KernelUCB [1]:** At every round, we have $N$ arms where the expected reward of each arm is fully characterized by its context variable. The arms have *different* context variables at each round. We learn *one shared* set of weights to predict the expected reward for the $N$ observed contexts (i.e., arms) in the next round.
>
> As explained above, in the KernelUCB [1] setting, there is not a fixed model corresponding to one arm across iterations, and the arms will perform independently across iterations depending on their context. However, in the setting of PAK-UCB (our method), each arm will represent one fixed generative model in all the learning rounds.
>
> [1] Valko, et al. "Finite-time analysis of kernelised contextual bandits", UAI 2013.

---

### Official Review · Reviewer_MgCP · 2025-03-20

**Overall Recommendation:** 3

**Summary:**

This study focuses on the task of selecting the generative model that achieves the highest reward for a given input prompt. The authors formulate this task as a contextual bandit (CB) problem, treating it as an online learning problem where past records are used to update the predictive model dynamically. They explore a UCB-based approach to solve the CB problem, specifically introducing PAK-UCB, which employs a kernel-based prediction function for each model (arm). Additionally, they present RFF-UCB, which reduces computational burden at the cost of some performance loss. The proposed methods were validated on the text-to-image generation task using models such as Stable Diffusion v1.5 and PixArt-alpha, demonstrating superior performance compared to other UCB-based approaches.

**Claims And Evidence:**

This study argues that the performance of generative models can vary significantly across different prompt categories. This claim is supported by the analysis in Figure 1 and further reinforced by the main experimental results, which show that approaches selecting the optimal model for each prompt outperform the strategy of consistently choosing a generally high-performing model (One-arm Oracle).

**Essential References Not Discussed:**

This study explores UCB-based approaches for the task of model selection. However, there may be alternative methodologies for this task beyond UCB. Notably, as presented in [1], an Agent AI approach is also a viable option. It would be beneficial to include a discussion on alternative methodologies beyond UCB as well.

[1] DiffusionGPT: LLM-Driven Text-to-Image Generation System

**Experimental Designs Or Analyses:**

I believe that the overall evaluation setting, including the metrics and baselines, is well-designed. However, it is unclear which prompt set was used for the main experiments presented in the paper. Based on inferences from Appendix Section 7, Figure 10, and Figure 12, it seems that the experiments were conducted using only two categories from the MS-COCO dataset. If that is the case, it would be beneficial to validate the approach not only on such a clearly defined and limited category set but also on more general prompt sets, such as ImageRewardDB[1], for a more comprehensive evaluation.

[1] ImageReward: Learning and Evaluating Human Preferences for Text-to-Image Generation

**Methods And Evaluation Criteria:**

In this study, the authors formulate the given task as a contextual bandit (CB) problem, which I believe is a valid problem formulation from the perspective of online model selection. Additionally, their proposed method (PAK-UCB), which defines a different kernel function for each arm, is also a reasonable approach within the CB framework.

However, regarding this methodology, the authors emphasize the use of an "arm-specific" prediction function as a key distinguishing feature compared to other UCB-based approaches. However, based on my understanding, using an arm-specific prediction function is a commonly adopted approach in previous CB-related studies [1], making this claim somewhat overemphasized. Therefore, it would be beneficial to compare their method with other UCB approaches that also utilize arm-specific functions and discuss the differences between them.

[1] A Contextual-Bandit Approach to Personalized News Article Recommendation

**Other Comments Or Suggestions:**

While this study focuses on selecting a model based on the input prompt, another practical approach involves selecting an appropriate fine-tuned adapter for each prompt, as proposed in [1]. I am curious whether the proposed PAK-UCB method could also be applied to such scenarios, where adapters are chosen based on the prompt. Unlike models, adapters are generally more abundant (typically at least 10 or more), so demonstrating the effectiveness of the proposed method in cases where the contextual bandit (CB) has more than three arms could further highlight its generalizability.

Additionally, if the concerns mentioned above are addressed, I would be willing to increase the rating.

[1] Stylus: Automatic Adapter Selection for Diffusion Models

**Other Strengths And Weaknesses:**

Strengths: Identifying the importance of generative model selection based on prompts and formulating it as a contextual bandit problem is novel and interesting. Additionally, the presentation is clear and well-structured.

Weaknesses: Achieving meaningful performance requires multiple iterations, and even after several iterations, the model seems to work effectively only when the prompts fall within previously seen categories. While the appendix (Fig 12) demonstrates adaptation ability through additional training when a new category is introduced, handling unseen categories still necessitates substantial additional learning, which may limit the practical usability of the proposed approach.

**Questions For Authors:**

As mentioned in the "Experimental Designs or Analyses" section, it would be helpful to specify the exact prompt set used in the experiments presented in the main paper.

**Relation To Broader Scientific Literature:**

The "generative model selection based on prompts" emphasized in this study is a valuable research topic that could be further explored in future work. Additionally, this study has identified a meaningful domain where UCB literature can be effectively applied.

**Theoretical Claims:**

In this study, the authors theoretically derive the regret bound for PAK-UCB. Since they appropriately extend the proofs used in UCB-based approaches, I find this approach valid and well-founded.

---

> ### Author Rebuttal · Authors · 2025-03-31
>
> We thank Reviewer MgCP for the thoughtful feedback on our work. Below is our answer to the reviewer's comments and questions.
>
> **1. Arm-specific reward model**
>
> We thank the reviewer for pointing out the bandit algorithm in [1] that utilizes arm-specific prediction functions. To the best of our knowledge, the arm-specific bandit algorithms in the literature (References [1-3]) consider a linear pay-off. However, in our experiments, we observed that the PAK-UCB algorithm performs better when non-linear kernel functions (e.g., polynomial kernel and RBF kernel) are applied. We think this phenomenon is due to the normalized output of the CLIP embedding that is on the surface of the unit sphere, where a linear classification could be sub-optimal compared to a non-linear rule provided by the kernel-based approach. In the revision, we will discuss the existing arm-specific implementations of LinUCB and the necessity of including non-linear kernel functions to obtain better results in the case of text-to-image model selection.
>
> [1] Li et al. "A contextual-bandit approach to personalized news article recommendation." WWW 2010.
>
> [2] Fang et al. "Networked bandits with disjoint linear payoffs." KDD 2014.
>
> [3] Xu et al. "Contextual-bandit based personalized recommendation with time-varying user interests." AAAI 2020.
>
>
> **2. Performance on ImageReward DB dataset**
>
> Thank you for the suggestion. Due to the limited time, we report preliminary results on the ImageRewardDB ReFL dataset [1]. In the experiment, the algorithm selects among three T2I models considered in our paper: StableDiffusion v1.5, UniDiffuser, and PixArt-$\alpha$, which attain a CLIPScore of 35.54, 34.40, and 37.20 on (a subset of) the dataset, respectively. After 5k iterations, our proposed PAK-UCB-poly3 and RFF-UCB algorithms attain an OPR (ratio of picking the best model PixArt-$\alpha$) of 63.76% and 42.94%. On the other hand, the KernelUCB-poly3 baseline achieves only 34.18%. We will cite and discuss the dataset in the revised paper.
>
> [1] Xu et al. "Imagereward: Learning and evaluating human preferences for text-to-image generation." NeurIPS 2023.
>
>
> **3. Comparison with the DiffusionGPT framework [1]**
>
> We thank the reviewer for pointing out the alternative DiffusionGPT method in [1], which leverages an LLM agent for model selection. Our proposed PAK-UCB approach can be viewed as complementary to the DifussionGPT method, which employs the UCB bandit approach to address this task. We will cite and discuss the potential combination of the PAK-UCB and DiffusionGPT approaches as a future direction in the revised text.
>
> [1] Qin et al. "DiffusionGPT: LLM-driven text-to-image generation system."
>
>
> **4. Handling unseen categories**
>
> We agree with the reviewer that adaptation to unseen prompt categories would be challenging if the new prompts were *fully orthogonal* to previous prompts. On the other hand, in practice, it is often the case that the incoming prompt has *some correlation* with some previous prompts. For example, a user who has generated images of cats and dogs is likely to generate images of other pets or animals in the future. Assuming that the optimal model choice changes continuously with input prompts, the PAK-UCB online learning approach would be capable of predicting the optimal arm after observing previously correlated prompts.
>
> To test this hypothesis, we conducted a numerical experiment to predict the CLIPScore of StableDiffusion v1.5. We train a poly3-based prediction model on $n=1,2,3$ categories and compute the prediction MSE on a new category after observing 10 samples in this new category. The results show that the prediction function can generalize effectively to unseen but correlated prompt categories.
>
> |Trained categories|bird|bird, horse|bird, horse, sheep|
> |-|-|-|-|
> |**New category**|horse|sheep|cow|
> |**Prediction MSE**|$0.22$|$0.19$|$0.08$|
>
>
> **5. Performance on a large number of arms**
>
> We appreciate the reviewer's comment on setups with a large number of arms. Please note that the primary goal of the online selection task is to minimize the regret. Our regret bound in Theorems 1 and 2 are on the order of $\widetilde{O}(\sqrt{GT})$ for $G$ arms and time horizon $T$. In practice, the effective number of arms will be lower if the performance scores change more smoothly across arms.
>
> To numerically test the effect of a larger number of arms, we conducted an experiment to evaluate PAK-UCB-poly3 and RFF-UCB in a setting with ten arms: We included five more arms to the synthetic T2I task (Setup 3), which generate a clean image less frequently (only $10\%$ of the time). The results show that PAK-UCB-poly3  and RFF-UCB can outperform the best single arm and the KernelUCB baseline.
>
> |Metric (After 2k iterations)|PAK-UCB-poly3|RFF-UCB|KernelUCB-poly3|
> |-|-|-|-|
> |**O2B**|0.84|2.05|0.22|
> |**OPR**|0.13|0.16|0.11|

---

> > ### Comment · Reviewer_MgCP · 2025-04-03
> >
> > Thanks to the authors for their rebuttal. It addressed most of my concerns, so I have increased the score. It would be great if the authors could include the experimental results on unseen prompts and various prompt sets mentioned in the rebuttal.

---

> > > ### Author Response · Authors · 2025-04-04
> > >
> > > We sincerely thank Reviewer MgCP for the constructive suggestions and the feedback on our response. We are glad to hear that our responses could address the concerns. We will include the discussed numerical results in the revised paper.

---

### Decision · Program_Chairs · 2025-05-01

**Decision:**

Accept (poster)

**Comment:**

This paper formulates model selection as a bandit problem, specifically designing a contextual bandit algorithm named PAK-UCB and proving upper bounds on its expected regret. Additionally, the authors introduce an approximate version of PAK-UCB tailored for practical applications. The effectiveness of the proposed algorithms is verified through applications in text-to-image and image captioning tasks.

The problem addressed is well-motivated, and the novel variant of the contextual bandit problem is intriguing. The experimental results are also robust. However, the connection between the proposed bandit formulation and the demonstrated application domains (mainly in image generation) is not entirely clear. For instance, extending the experiments to include other generative models, such as LLMs, could enhance the paper’s impact. Nonetheless, I believe that accepting this work would be a valuable contribution to the community. Therefore, I recommend a weak accept.